# Conduction band convergence and local structure distortion for superior thermoelectric performance of GaSb-doped n-type PbSe thermoelectrics

Jing Zhou [1,2,10], Hong-Hua Cui[3,10], Yukun Liu[4,10], Hongwei Ming[2], Yan Yu[1,2], Vinayak P. Dravid [4], Zhong-Zhen Luo [1,2,5] ✉, Qingyu Yan [6] ✉, Zhigang Zou [1,2,7,8] & Mercouri G. Kanatzidis [4,9] ✉

Achieving high-stability thermoelectric materials with excellent average power factor and figure of merit is crucial for maximizing the output power density and conversion efficiency of thermoelectric devices. In this study, GaSb is added to PbSe as an n-type dopant to form stable solid solutions. Doping with GaSb flattens the conduction band and reduces the energy difference between the $\Sigma$ and $L$ conduction bands, thereby significantly improving the Seebeck coefficient. Herein, the Ga and Sb atoms co-occupy the vacant Pb sites, unlike in the case of traditional single-element doping, as is verified by density functional theory calculations. The resultant structural distortion is confirmed via transmission electron microscopy. This local structure distortion caused by GaSb doping reduces the lattice thermal conductivity. Consequently, the $Pb_{0.99875}(GaSb)_{0.00125}Se$ sample exhibits a record-high average power factor of ~22.37 $\mu W\,cm^{-1}\,K^{-2}$ and a high average figure of merit of ~0.94 in the temperature range of 300–873 K. Furthermore, the introduction of interstitial Cu and discordant Zn atoms further reduces the lattice thermal conductivity. The $Pb_{0.99875}(GaSb)_{0.00125}Zn_{0.01}Se_{1.01}$-0.3%Cu sample exhibits a low lattice thermal conductivity of ~0.4 $W\,m^{-1}\,K^{-1}$ at 873 K and a record-high average figure of merit of ~1.01 in the temperature range of 300–873 K.

Thermoelectric materials are functional materials that can directly interconvert heat and electricity[1–7]. They play an irreplaceable role in waste heat power generation and solid-state refrigeration[8–11]. Thermoelectric devices comprising n- and p-type thermoelectric materials have small volumes, emit no pollutants, generate no noise, have a wide applicable temperature range, and are highly reliable. The energy conversion efficiency of thermoelectric materials depends on the dimensionless figure of merit, $ZT = S^2\sigma T/(\kappa_{ele} + \kappa_{lat})$, where $S$ is the Seebeck coefficient, $\sigma$ is the electrical conductivity, $T$ is the absolute temperature, and $\kappa_{ele}$ and $\kappa_{lat}$ represent the charge carrier and lattice thermal conductivities, respectively[12–16]. The quantity $S^2\sigma$ is defined as the power factor ($PF$), which determines the electrical transport performance of thermoelectric materials[17–22]. The performance of a thermoelectric device is evaluated based on its thermoelectric conversion efficiency ($\eta$), which is determined using the formula: $\eta = [(T_H - T_C)/T_H][(1 + ZT_{avg})^{1/2} - 1]/[(1 + ZT_{avg})^{1/2} + T_C/T_H]$, and from its output power density ($\omega$), which is calculated using the formula: $\omega = [(T_H - T_C)/4L]PF_{avg}$[23–26], where $T_H$ and $T_C$ are the hot-side and cold-side temperatures and $L$ is the length of the thermoelectric leg in the device. $PF_{avg}$ and $ZT_{avg}$ are the average $PF$ and $ZT$ values over the temperature range

from $T_C$ to $T_H$. The simultaneous achievement of a high $PF_{avg}$ and $ZT_{avg}$ is crucial for the development of highly efficient devices[27–29].

PbTe, which is an excellent mid-temperature range thermoelectric material, has been extensively studied. It has achieved a $ZT_{avg} > 1$ in various compositions, such as $(PbTe)_{81}Sb_2Te_3$-0.6Sb-$2Cu_2Te$[30], $Pb_{0.975}Ga_{0.025}Te$-0.25%ZnTe[31], $Pb_{0.98}Sb_{0.02}Te$[5], and $Pb_{1.01}Te_{0.998}I_{0.002}$-0.002Ag[32]. However, its widespread application is limited owing to the low abundance of Te (only 0.001 ppm) in the Earth's crust[33]. PbSe is considered a promising alternative owing to its similar crystal and electronic structures to PbTe, with Se being approximately 50 times more abundant than Te[34,35]. However, compared to PbTe, PbSe has a smaller bandgap, a lower Seebeck coefficient, and higher thermal conductivity, resulting in an inferior thermoelectric performance.

Recent studies have reported the improved thermoelectric performance for the n-type PbSe with a reduced $\kappa_{lat}$ via all-scale hierarchical architectures[36–39]. $\kappa_{lat}$ has nearly reached the limit for amorphous materials. However, all-scale hierarchical architectures inevitably aggravate carrier scattering, thus reducing carrier mobility ($\mu_H$)[36,40–42]. With limited scope for further $\kappa_{lat}$ reduction, enhancement of $PF$ is crucial for achieving higher $ZT$ and $\omega$. Compared with p-type PbSe, the large energy difference ($\Delta E_c$) between the first ($L$) and second ($\Sigma$) conduction bands poses a significant challenge for achieving conduction band convergence and a high $PF$ in the n-type PbSe[33].

A high $\mu_H$ is crucial for improving the thermoelectric performance over a broad temperature range. Undoped PbSe possesses Pb vacancies, leading to a low $\mu_H$. Lattice planification strategies have been widely used to improve the $\mu_H$ of PbSe-based thermoelectric materials[43,44]. We hypothesized that introducing a few compounds with high $\mu_H$ into the PbSe matrix without forming a secondary phase to achieve lattice planarization could result in a high $\mu_H$ in the as-formed solid solution. Most III-V semiconductors have high $\mu_H$, such as GaAs ($\mu_H$ =~ 4400 $cm^2 V^{-1} s^{-1}$) and InP ($\mu_H$ = ~4600 $cm^2 V^{-1} s^{-1}$)[45]. Among them, those with high $\mu_H$ and narrow bandgap are often selected as dopants to optimize the thermoelectric properties of PbSe.

In this study, we hypothesized that doping n-type PbSe with GaSb, a semiconductor with high $\mu_H$, could enhance both the $PF_{avg}$ and $ZT_{avg}$ by leveraging the unique co-occupancy of Ga and Sb atoms on Pb sites in the PbSe matrix. This co-occupancy was expected to create an n-type solid solution with superior thermoelectric performance by optimizing carrier concentration and reducing $\kappa_{lat}$. To further enhance the thermoelectric properties, we hypothesized that the addition of interstitial Cu and discordant Zn atoms could disrupt lattice phonon transport, thereby minimizing $\kappa_{lat}$. We achieved a high $PF$ value of ~32 $\mu W\,cm^{-1}\,K^{-2}$ and a large $ZT$ value of ~0.37 at room temperature. Remarkably, a record-high $PF_{avg}$ of ~22.37 $\mu W\,cm^{-1}\,K^{-2}$ was achieved in the temperature range of 300–873 K. The introduction of Cu and Zn reduced the $\kappa_{lat}$ to ~0.4 $W\,m^{-1}\,K^{-1}$ at 873 K, culminating in a high $ZT_{avg}$ of ~1.01 in the same temperature range. These results highlight the effectiveness of our doping strategy in simultaneously optimizing electronic and thermal transport properties in PbSe-based thermoelectric materials.

## Results

### Structural characterization

The PXRD patterns of the $Pb_{1-x}(GaSb)_xSe$ ($x$ = 0, 0.05%, 0.075%, 0.1%, 0.125%, 0.15%, and 0.175%) samples shown in Fig. 1a confirm that all samples are single-phase and crystallized in the rock-salt structure with the $Fm\bar{3}m$ space group (PDF#06-0354). As shown in Fig. 1b, the lattice parameters of the GaSb-doped PbSe increased with increasing GaSb doping amount. This can be attributed to the replacement of Pb with Ga and Sb.

### Micro- and nano-structure analyses

The microstructure of the $Pb_{0.99875}(GaSb)_{0.00125}Se$ sample was analyzed. Figure 1c shows the backscattered electron (BSE) SEM image, which exhibits a uniform contrast, indicating the absence of any noticeable second-phase region. Figure 1d shows the energy dispersive spectroscopy (EDS) results, demonstrating that Pb, Se, Ga, and Sb are evenly distributed within the $Pb_{0.99875}(GaSb)_{0.00125}Se$ sample.

### Microstructure

The occupation conditions of GaSb in PbSe were analyzed using density functional theory (DFT). A detailed discussion of the results is presented in the methods section. The DFT results suggest that the Ga and Sb atoms co-occupy vacant Pb sites in GaSb semiconductor

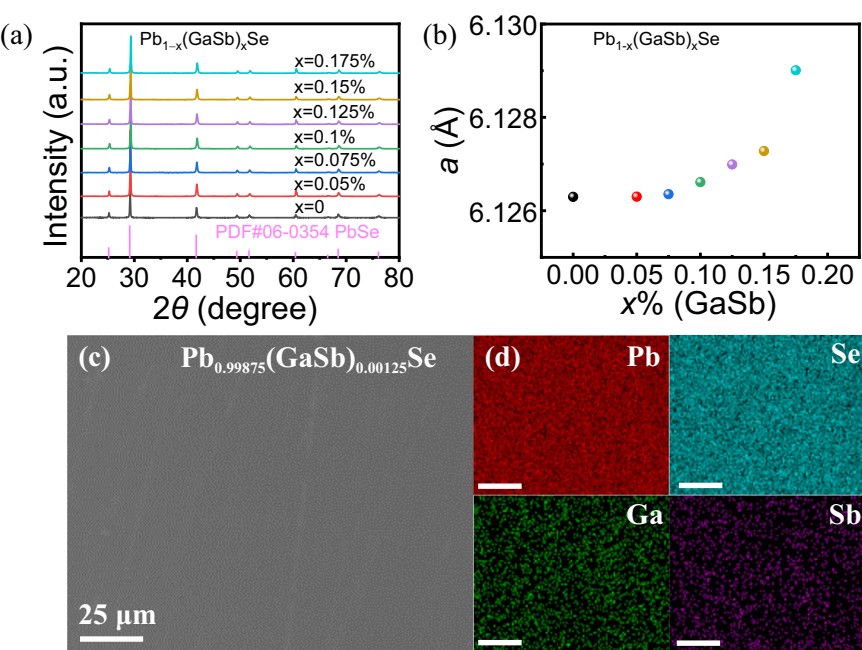

**Fig. 1 | Phase characterization of $Pb_{1-x}(GaSb)_xSe$ samples. a** PXRD patterns and **b** refined lattice parameters of the $Pb_{1-x}(GaSb)_xSe$ ($x$ = 0, 0.05%, 0.075%, 0.1%, 0.125%, 0.15%, and 0.175%) samples as a function of the GaSb content. **c** SEM-BSE image of the $Pb_{0.99875}(GaSb)_{0.00125}Se$ sample and **d** EDS mapping of the region shown in **c**.

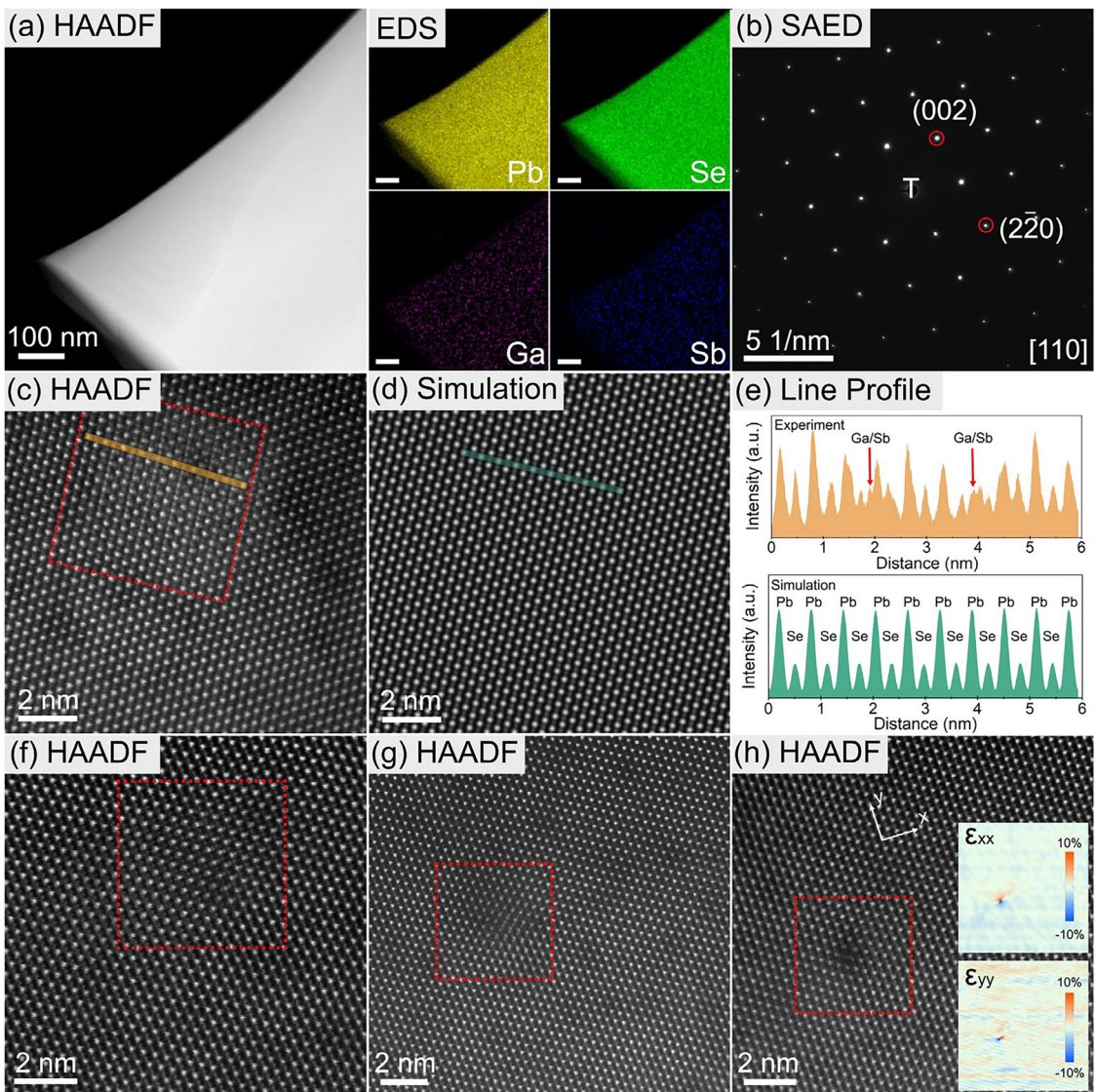

**Fig. 2 | S/TEM analysis of the microstructure of the $Pb_{0.99875}(GaSb)_{0.00125}Se$ sample. a** HAADF image of a representative region with the corresponding EDS maps, where all elements are homogeneously distributed. Weak signals of Ga and Sb are observed owing to a low doping concentration. **b** SAED pattern along the [110] zone axis acquired from the region shown in (**a**). **c** Atomic-resolution HAADF image showing the local structural distortion. **d** Multislice HAADF image simulation of pure PbSe without structural distortion. **e** Intensity line profiles of the regions highlighted in (**c**, **d**). **f**, **g** HAADF images of representative local structural distortions in the microstructure. **h** HAADF image of an edge dislocation induced by local structural distortion. The strain distribution is shown in the inset image: $y = [2\bar{2}0]$ and $x = [00\bar{2}]$.

doping, unlike in the case of a Ga atom or an Sb atom occupying a Pb position. Therefore, the microstructure of the $Pb_{0.99875}(GaSb)_{0.00125}Se$ sample was examined using S/TEM. Figure 2a shows the high-angle annular dark-field (HAADF) image of a representative region of the $Pb_{0.99875}(GaSb)_{0.00125}Se$ sample. The uniform contrast in the low-magnification HAADF image indicates the absence of phase segregation or precipitate formation. This observation is supported by EDS, which revealed a uniform distribution of the elemental species. Notably, the GaSb doping concentration was low and close to the detection limit, which resulted in weak EDS signals for Ga and Sb. In addition, the selected area electron diffraction (SAED) pattern (Fig. 2b) along the [110] axis from the region shown in Fig. 2a matches the rock-salt structure with the space group $Fm\bar{3}m$. No additional diffraction spots or streaking were observed in the SAED pattern, suggesting that the material is primarily a solid solution. However, the atomic-resolution HAADF images revealed local structural distortions in the microstructure. Figure 2c presents an atomic-resolution HAADF image acquired along the [110] zone axis, which demonstrates alternating projections of cationic and anionic columns. This clearly indicates that, in addition to the periodic alternating cation and anion columns, additional atoms with smaller sizes are present. For comparison, a HAADF image of pure PbSe was simulated using the multislice method (details are provided in the Experimental Section), as shown in Fig. 2d. Figure 2e shows the intensity line profiles extracted from the regions highlighted in Fig. 2c, d. The intensity line profiles from the simulated results showed spatially uniform contrast alternations originating from the different atomic numbers of Pb ($Z = 82$) and Se ($Z = 34$). In contrast, experimental results revealed a decrease in cation column intensity, accompanied by the appearance of additional atomic columns nearby. The cation columns exhibiting reduced intensity also displayed smaller atomic radii. Additional atoms corresponding to Ga ($Z = 31$) and Sb ($Z = 51$) were observed near Pb sites, exhibiting significantly lower intensities than Pb in the HAADF image.

To investigate the origin of these image feature changes, we performed simulation-based analysis to understand how different local atomic configurations influence the observed contrasts and to

evaluate the likelihood of GaSb co-occupying the deficient Pb site, as predicted by DFT. When GaSb co-doping occurs at the Pb site, multiple configurations are possible. Here, we consider two representative and simplified cases to analyze how co-doping alters the image features.

In the first configuration, Ga occupies the deficient Pb site, while Sb, unable to fit within the site, shifts by 1/3 of the unit cell along the [010] direction (Atomic Model 1, Fig. S1). To emphasize contrast changes, we assume this co-occupation extends uniformly along the z-direction. Multislice simulations suggest that the resulting local lattice distortion closely resembles experimental observations, as highlighted by red arrows in Fig. S2a. The line profiles indicate a decrease in intensity at cation sites due to Ga's significantly lower atomic number ($Z = 31$) compared to Pb ($Z = 82$). Additionally, cation column radii decrease where co-doping occurs, as Ga (atomic radius = 130 pm) is smaller than Pb (180 pm). Moreover, the line profile along the [010] direction reveals an additional atomic column overlapping with the Se columns (Fig. S3a), arising from Sb's displacement. These simulated results align well with experimental data, further confirming the structural distortions.

In the second configuration, Sb occupies the Pb site, while Ga, unable to co-occupy the same site, shifts by 1/3 of the unit cell along the [010] direction (Atomic Model 2, Fig. S1). Again, we assume uniform extension along the z-direction. Line profiles indicate a similar intensity drop at cation sites, this time due to Sb's lower atomic number ($Z = 51$) compared to Pb ($Z = 82$). As in the first case, the projected cation column radii decrease in co-doped regions, as Sb (145 pm) remains smaller than Pb (180 pm). The line profile along the [010] direction again reveals an additional atomic column overlapping with the Se columns (Fig. S3b), consistent with Ga's displacement. These simulation results match experimental observations (Fig. S2a), reinforcing the co-doping mechanism.

Thus, GaSb co-occupation at the deficient Pb site leads to a local decrease in image intensity at cation sites and a reduction in atomic column radii, irrespective of whether Ga or Sb directly occupies the Pb site. Furthermore, if the second atom in the GaSb pair cannot fit within the Pb site, an additional atomic column becomes visible. This serves as key evidence of GaSb co-doping without dissociation. Although the precise atomic configuration remains ambiguous due to HAADF imaging capturing projected atomic arrangements, the overall features align with the predicted trends. The experimentally observed local lattice distortions are consistent with these simulations, further supporting GaSb co-occupation at the deficient Pb site, as corroborated by DFT calculations.

Because of the low doping concentration of GaSb, these structural distortions are highly localized and have a low density of distribution, rendering them invisible in the low-magnification HAADF images and SAED pattern (Fig. 2a, b, respectively). Fig. 2f, g shows the different atomic configurations of the structural distortions in the microstructure, demonstrating the general existence of these distortions. In particular, some configurations can lead to the formation of edge dislocations (Fig. 2h). Based on geometric phase analysis (GPA), additional strain fluctuation was introduced to suppress phonon propagation in addition to the local lattice mismatch due to structural distortion. Previously, we showed that off-center discordant Ge atoms in the PbTe matrix cause intense phonon scattering[46,47].

## Hall effect

The temperature-dependent Hall coefficients of the Pb$_{1-x}$(GaSb)$_x$Se ($x = 0$, 0.05%, 0.075%, 0.1%, 0.125%, 0.15%, and 0.175%) samples were measured to understand the effect of GaSb doping on their charge transport properties. As shown in Fig. 3a, the absolute value of the Hall coefficient ($R_H$) decreased with increasing GaSb doping amount. The carrier concentration ($n_H$) increased from $0.99 \times 10^{19}$ cm$^{-3}$ for $x = 0$ to $4.79 \times 10^{19}$ cm$^{-3}$ for $x = 0.175\%$ at room temperature, as shown in Fig. 3b.

As shown in Fig. 3c, the carrier mobility ($\mu_H$) of all GaSb-doped samples exceeded 300 cm$^2$ V$^{-1}$ s$^{-1}$ at 300 K. With increasing GaSb content, $\mu_H$ decreased from 527 cm$^2$ V$^{-1}$ s$^{-1}$ for $x = 0.05\%$ to 408 cm$^2$ V$^{-1}$ s$^{-1}$ for $x = 0.175\%$ at 300 K, which was primarily due to an increase in $n_H$. Figure 3d shows the relationship between $n_H$ and $\mu_H$; the Pb$_{0.99875}$(GaSb)$_{0.00125}$Se sample exhibits a relatively higher $\mu_H$ than the other high-performance n-type PbSe-based materials, such as PbSe(Gd/Br)[48], PbSe(Cd)[49], PbSe(Ge/Sb)[37], PbSe(In)[38], PbSe(Ga)[50], and PbSe(Br)[51]. The PbSe(Br)-based samples exhibited an ultrahigh $\mu_H$ because Br is closest in atomic size and electronic structure to Se, thus minimally affecting the $\mu_H$. Unlike heavy-element doping, a small amount of GaSb doping can effectively reduce the scattering effect of Pb vacancies on the carriers. Concurrently, a stable solid solution is formed without the introduction of second-phase impurities, which is also conducive to obtaining a high $\mu_H$. Therefore, the Pb$_{0.99875}$(GaSb)$_{0.00125}$Se sample exhibited a high $\mu_H$.

## Electrical conductivity and Seebeck coefficients

The temperature-dependent electrical conductivity of the Pb$_{1-x}$(GaSb)$_x$Se ($x = 0$, 0.05%, 0.075%, 0.1%, 0.125%, 0.15%, and 0.175%) samples increased significantly with increasing GaSb content, reaching a maximum value of ~3180 S cm$^{-1}$ at 300 K for the Pb$_{0.99825}$(GaSb)$_{0.00175}$Se sample, as shown in Fig. 3e. The electrical conductivity decreased with increasing temperature for all samples, indicating their degenerative semiconducting behavior. This improved electrical conductivity was attributed to the increase in $n_H$ with increasing GaSb content.

Figure 3f presents the temperature-dependent Seebeck coefficients for the Pb$_{1-x}$(GaSb)$_x$Se ($x = 0$, 0.05%, 0.075%, 0.1%, 0.125%, 0.15%, and 0.175%) samples. The negative Seebeck coefficients indicated that the samples were n-type, with electrons as the dominant charge carriers across the entire temperature range. For the GaSb-doped samples, the Seebeck coefficient decreased from −174 to −72 μV K$^{-1}$ as the GaSb content increased from 0.05% to 0.175% at 300 K. This behavior is consistent with the dependence of the Seebeck coefficient on $n_H$. The Pisarenko relation plot for the Pb$_{1-x}$(GaSb)$_x$Se ($x = 0$, 0.05%, 0.075%, 0.1%, 0.125%, 0.15%, and 0.175%) samples at 300 K was constructed (Fig. 3g) to further investigate this relationship. The red curve represents the calculated values for n-type PbSe-based materials according to the single parabolic band (SPB) model with a density of states (DOS) effective mass of $0.53m_e$ for electrons, where $m_e$ is the free electron mass. GaSb doping was found to increase the DOS effective mass from $0.43m_e$ to $0.53m_e$ in PbSe, leading to a higher Seebeck coefficient for the Pb$_{1-x}$(GaSb)$_x$Se ($x = 0.05\%$, 0.075%, 0.1%, 0.125%, 0.15%, and 0.175%) samples than for the other high-performance n-type PbSe-based materials. Figure 3h shows the Pisarenko relation plots for the Pb$_{0.99875}$(GaSb)$_{0.00125}$Se sample at different temperatures (300, 373, 473, 573, 673, and 773 K). The DOS effective mass increased from $0.53m_e$ to $0.86m_e$ as the temperature increased from 300 to 773 K.

## Power factor

The temperature-dependent $PF$ values of the Pb$_{1-x}$(GaSb)$_x$Se ($x = 0$, 0.05%, 0.075%, 0.1%, 0.125%, 0.15%, and 0.175%) samples were calculated based on the measured electrical conductivity and Seebeck coefficient values, as shown in Fig. 3i. The $PF$ values of the GaSb-doped samples increased significantly owing to the enhanced $n_H$ while maintaining a high $\mu_H$. GaSb-doped samples exhibited $PF$ values greater than 15 μW cm$^{-1}$ K$^{-2}$ at 300 K, with the Pb$_{0.99875}$(GaSb)$_{0.00125}$Se sample exhibiting the highest $PF$ value of ~32 μW cm$^{-1}$ K$^{-2}$ at 300 K.

## DFT calculations of the electronic structure

We calculated the electronic band structure of GaSb-doped PbSe-based material to investigate the reason for its high $PF$ value. Figure 4a, c

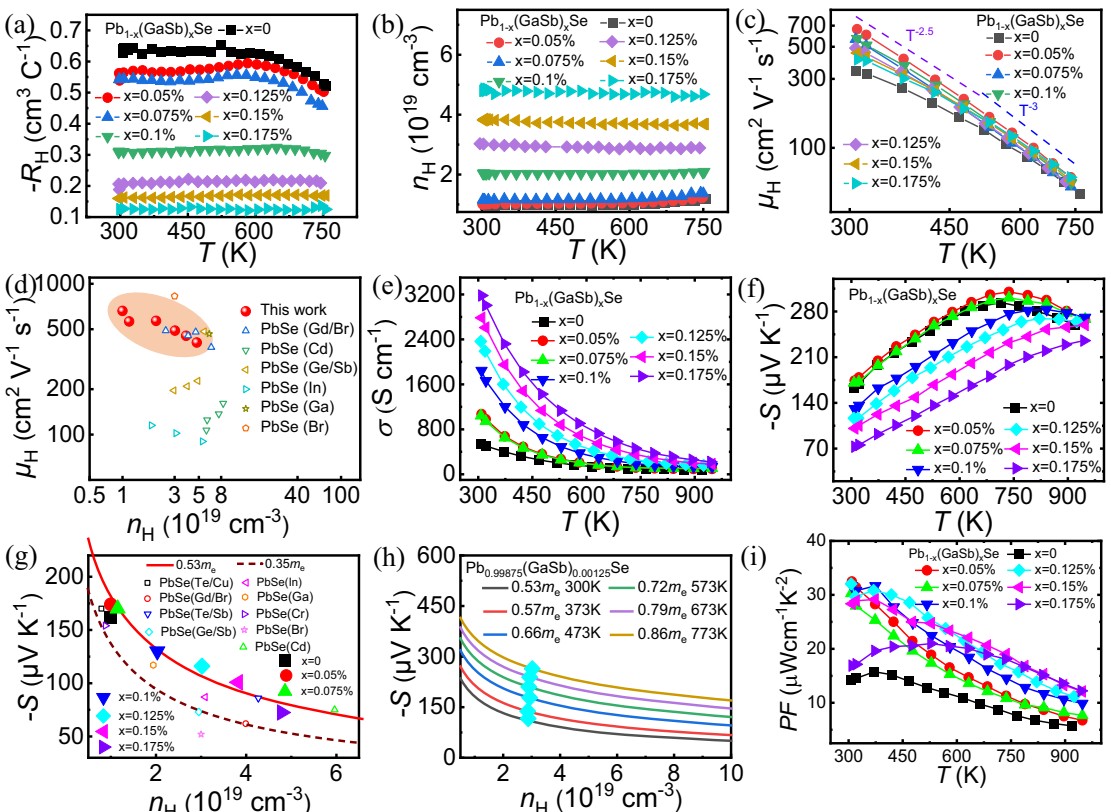

**Fig. 3 | Electrical transport properties of Pb$_{1-x}$(GaSb)$_x$Se samples.** Temperature-dependent **a** hall coefficient, $R_H$; **b** carrier concentration, $n_H$; and **c** carrier mobility, $\mu_H$ for the Pb$_{1-x}$(GaSb)$_x$Se ($x$ = 0, 0.05%, 0.075%, 0.1%, 0.125%, 0.15%, and 0.175%) samples. **d** $\mu_H$ as a function of $n_H$ for n-type PbSe-based thermoelectrics. Temperature-dependent **e** electrical conductivity, $\sigma$, and **f** Seebeck coefficient, $S$ for the Pb$_{1-x}$(GaSb)$_x$Se ($x$ = 0, 0.05%, 0.075%, 0.1%, 0.125%, 0.15%, and 0.175%) samples. **g** $S$ as a function of $n_H$ at room temperature for n-type PbSe-based thermoelectrics. **h** $S$ as a function of $n_H$ at different temperatures (300, 373, 473, 573, 673, and 773 K) for the Pb$_{0.99875}$(GaSb)$_{0.00125}$Se sample. **i** Power factor, $PF$ as a function of temperature for Pb$_{1-x}$(GaSb)$_x$Se ($x$ = 0, 0.05%, 0.075%, 0.1%, 0.125%, 0.15%, and 0.175%) samples.

show the electronic band structures of the pure and GaSb-doped PbSe-based material, respectively. The conduction bands of the GaSb-doped PbSe-based material were significantly different from those of the pure PbSe-based material (Fig. S4). The conduction band of GaSb-doped PbSe-based material crosses the Fermi level, mainly because the $n_H$ increases due to the doping of GaSb, which makes the carrier fill the conduction band. In particular, the conduction band of the GaSb-doped PbSe-based material was flatter, which increased the DOS effective mass. This is conducive to obtaining a large Seebeck coefficient. In addition, GaSb doping decreased the $\Delta E_c$ between the $\Sigma$ and $L$ bands from 0.33 eV for the intrinsic PbSe to 0.08 eV for the GaSb-doped PbSe-based material. This decrease in $\Delta E_c$, which was primarily because of the upward shift of the $L$ band (Fig. S4), allowed the second conduction band to act as an additional electron transport channel, thereby improving the overall electrical performance. Figure S5 shows that the 6p orbitals of Pb predominantly contribute to the $\Sigma$ and $L$ conduction bands. As shown in Fig. S6, the replacement of Pb with GaSb reduced the contribution of the 6p orbitals of Pb to the $\Sigma$ and $L$ conduction bands. The contributions of the 4s and 4p orbitals of Ga and the 5s and 5p orbitals of Sb to the $L$ conduction band were minimal, causing the $L$ band to shift upward in the GaSb-doped PbSe-based material. The 5p orbitals of Sb contributed substantially to the $\Sigma$ conduction band, thereby offsetting the decreased contribution of the Pb orbitals and keeping the position of the $\Sigma$ conduction band essentially unchanged. Thus, GaSb doping promotes conduction band convergence. Also, we have done theoretical calculations closer to the experimental scale. Figure S7 shows the band structure of Pb$_{124}$(GaSb)$_1$Se$_{125}$. GaSb doping decreased the $\Delta E_c$ between the $\Sigma$ and $L$

bands from 0.33 eV for the intrinsic PbSe to 0.13 eV for the GaSb-doped PbSe-based material. According to the theoretical calculation results, as the doping amount of GaSb increases, the $\Delta E_c$ becomes smaller, indicating that GaSb does have the effect of reducing the $\Delta E_c$. Therefore, GaSb doping can promote conduction band convergence.

**Thermal conductivity**

As shown in Fig. 5a, the total thermal conductivity, $\kappa_{tot}$, increased from 1.96 W m$^{-1}$ K$^{-1}$ for $x$ = 0 to a maximum of 3.1 W m$^{-1}$ K$^{-1}$ for $x$ = 0.175% at 300 K. This can be attributed to the enhanced charge carrier thermal conductivity, $\kappa_{ele}$, which was calculated using the Wiedemann–Franz law[52]: $\kappa_{ele} = L\sigma T$, where $L$ is the Lorenz number, $\sigma$ is the electrical conductivity, and $T$ is the absolute temperature. $L$ was determined from the experimentally measured Seebeck coefficients using the equation: $L = 1.5 + \exp[-|S|/116]$[53], as shown in Fig. S8. The $\kappa_{lat}$ values of all samples were calculated by subtracting the $\kappa_{ele}$ values from the $\kappa_{tot}$ ($\kappa_{lat} = \kappa_{tot} - \kappa_{ele}$) values, as shown in Fig. 5b. GaSb doping led to a decrease in the $\kappa_{lat}$ values, and a further reduction was observed with increasing temperature. The Pb$_{0.99825}$(GaSb)$_{0.00175}$Se sample exhibited the lowest $\kappa_{lat}$ value (-1.09 W m$^{-1}$ K$^{-1}$) at 300 K (Fig. S9).

To further explore the effect of GaSb doping at the Pb sites, Ga-doped, Sb-doped, and Ga and Sb co-doped PbSe samples were prepared and compared. The phonon scattering contribution to $\kappa_{lat}$ was analyzed using the Debye–Callaway model[54,55]. U and N phonon–phonon process scattering, grain boundary scattering, and point defect scattering were mainly considered. As shown in Fig. 5c, P$_{Ga}$, P$_{Sb}$, P$_{Ga+Sb}$, and P$_{GaSb}$ represent the point defects in the Ga-doped, Sb-doped, Ga and Sb co-doped, and GaSb-doped PbSe-based

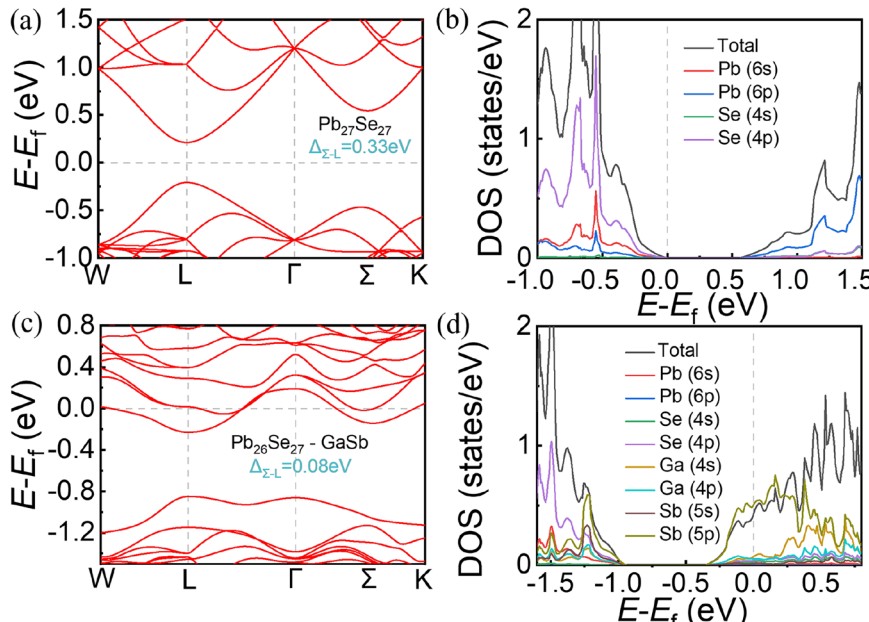

**Fig. 4 | Density functional theory calculations of the electronic structure. a** Electronic band structures and **b** projected DOS for the pure PbSe-based material. **c** Electronic band structures and **d** projected DOS for the GaSb-doped PbSe-based material.

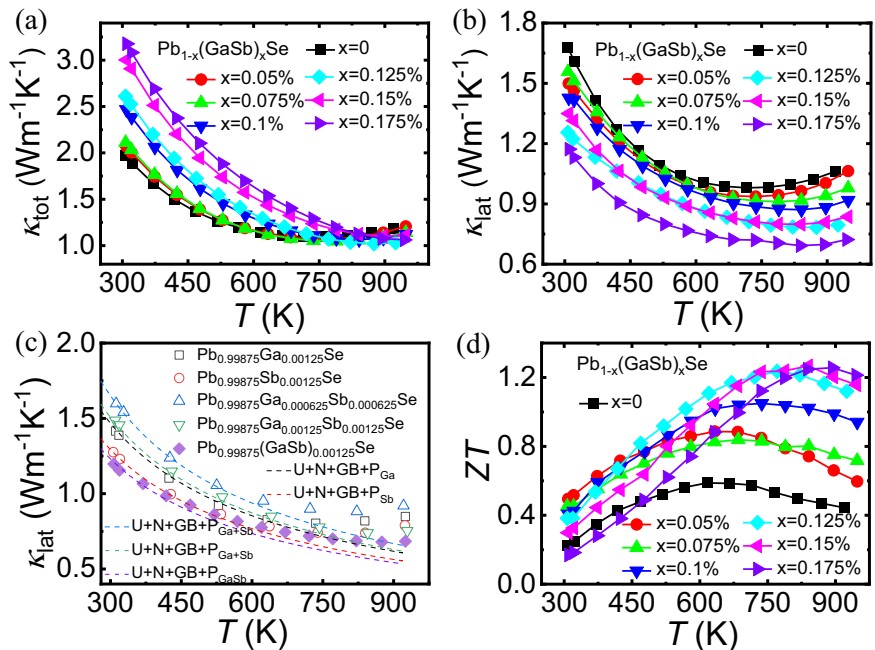

**Fig. 5 | Thermal conductivity as a function of temperature for the Pb$_{1-x}$(GaSb)$_x$Se ($x$ = 0, 0.05%, 0.075%, 0.1%, 0.125%, 0.15%, and 0.175%) samples. a** total ($\kappa_{tot}$) and **b** lattice ($\kappa_{lat}$) thermal conductivities. **c** Temperature-dependent lattice thermal conductivity ($\kappa_{lat}$) for Pb$_{0.99875}$Ga$_{0.00125}$Se, Pb$_{0.99875}$Sb$_{0.00125}$Se, Pb$_{0.99875}$Ga$_{0.000625}$Sb$_{0.000625}$Se, Pb$_{0.99875}$Ga$_{0.00125}$Sb$_{0.00125}$Se, and Pb$_{0.99875}$(GaSb)$_{0.00125}$Se samples. **d** Figure of merit ($ZT$) as a function of temperature for the Pb$_{1-x}$(GaSb)$_x$Se ($x$ = 0, 0.05%, 0.075%, 0.1%, 0.125%, 0.15%, and 0.175%) samples.

thermoelectric materials, respectively. The Pb$_{0.99875}$(GaSb)$_{0.00125}$Se sample exhibited the lowest $\kappa_{lat}$ value over the entire test temperature range, primarily because of local structural distortion and enhanced point defect scattering caused by GaSb doping at the Pb sites.

## Figure of merit

The temperature-dependent figures of merit ($ZT$) for the Pb$_{1-x}$(GaSb)$_x$Se ($x$ = 0, 0.05%, 0.075%, 0.1%, 0.125%, 0.15%, and 0.175%) samples are shown in Fig. 5d. The $ZT$ value of Pb$_{0.9995}$(GaSb)$_{0.0005}$Se

reached ~0.49 at 300 K owing to the high $PF$ value. With the co-optimization of the electrical and thermal transport performances, a high $PF$ value of ~32 μW cm$^{-1}$ K$^{-2}$ at 300 K and a low $\kappa_{lat}$ value of ~0.68 W m$^{-1}$ K$^{-1}$ at 773 K was achieved for the Pb$_{0.99875}$(GaSb)$_{0.00125}$Se sample. Consequently, it exhibited the maximum $ZT$ value of ~1.23 at 773 K.

## Addition of interstitial Cu atoms to decrease $\kappa_{lat}$

Cu was introduced as an interstitial atom in the Pb$_{0.99875}$(GaSb)$_{0.00125}$Se sample to further decrease its $\kappa_{lat}$ value. The

PXRD patterns of the $Pb_{0.99875}(GaSb)_{0.00125}Se$-yCu ($y = 0$, 0.1%, 0.3%, 0.5%, 0.7%, and 0.9%) samples shown in Fig. S10 confirm that all samples are single-phase. Meanwhile, the microstructure shows that the $Pb_{0.99875}(GaSb)_{0.00125}Se$-0.3%Cu sample is pure phase (Fig. S11). The dynamic doping effect of Cu ions generates extra charge carriers, optimizing $n_H$ over a wide temperature range and enhancing electrical conductivity at high temperatures. The temperature-dependent electrical conductivity of the $Pb_{0.99875}(GaSb)_{0.00125}Se$-yCu ($y = 0$, 0.1%, 0.3%, 0.5%, 0.7%, and 0.9%) samples increased with increasing Cu content (Fig. 6a). The electrical conductivity increased from 1240 to 2650 S cm$^{-1}$ as the Cu content increased from 0.1% to 0.9% at 300 K. The temperature-dependent Seebeck coefficients of the samples are presented in Fig. 6b. For the Cu-doped samples, the Seebeck coefficient decreased from −144 to −87 μV K$^{-1}$ as the Cu content increased from 0.1 to 0.9% at 300 K. The $\kappa_{lat}$ of all the Cu-doped samples decreased when the temperature exceeded 573 K (Fig. 6e). Lorenz number $L$ was determined from the experimentally measured Seebeck coefficients using the equation: $L = 1.5 + \exp[-|S|/116]$[53], as shown in Fig. S12. The $Pb_{0.99875}(GaSb)_{0.00125}Se$-0.3%Cu sample exhibited a low $\kappa_{lat}$ value of ~0.49 W m$^{-1}$ K$^{-1}$ at 723 K (Fig. 6e). Consequently, it achieved a high $ZT$ of ~1.41 at 823 K (Fig. 6f).

### Discordant Zn alloying to decrease $\kappa_{lat}$ further

The thermoelectric properties were further optimized via discordant Zn alloying. The PXRD patterns of the $Pb_{0.99875}(GaSb)_{0.00125}Zn_zSe_{1+z}$-0.3%Cu ($z = 0$, 0.125%, 0.25%, 0.5%, 1%, and 1.5%) samples shown in Fig. S13 confirm that all samples are single-phase. Meanwhile, the microstructure shows that the $Pb_{0.99875}(GaSb)_{0.00125}Zn_{0.01}Se_{1.01}$-0.3% Cu sample is pure phase (Fig. S14). The thermoelectric properties of the $Pb_{0.99875}(GaSb)_{0.00125}Zn_zSe_{1+z}$-0.3% Cu ($z = 0$, 0.125%, 0.25%, 0.5%, 1%, and 1.5%) samples as a function of temperature are shown in Fig. 7. The temperature-dependent electrical conductivity of the samples exhibited no significant change with varying Zn content at 300 K, except that for the $Pb_{0.99875}(GaSb)_{0.00125}Zn_{0.00125}Se_{1.00125}$-0.3% Cu samples, as shown in Fig. 7a. The temperature-dependent Seebeck coefficients of all the samples are shown in Fig. 7b. The absolute value of the Seebeck coefficient increased with increasing temperature. For the Zn-alloyed samples, Lorenz number $L$ was determined from the experimentally measured Seebeck coefficients using the equation: $L = 1.5 + \exp[-|S|/116]$[53], as shown in Fig. S15, the $\kappa_{tot}$ decreased from 0.98 to 0.82 W m$^{-1}$ K$^{-1}$ with increasing Zn content from 0 to 1% at 873 K (Fig. 7d). In the case of the $Pb_{0.99875}(GaSb)_{0.00125}Zn_{0.01}Se_{1.01}$-0.3% Cu sample, the $\kappa_{lat}$ value decreased over the entire test temperature range (Fig. 7e). It also exhibited a low $\kappa_{lat}$ value of ~0.4 W m$^{-1}$ K$^{-1}$ at 873 K, resulting in a high $ZT$ value of ~1.57 at the same temperature (Fig. 7f). The data with error bars is shown in Fig. S16.

$ZT_{avg}$ was calculated using the formula: $ZT_{avg} = \frac{1}{T_H - T_C} \int_{T_C}^{T_H} ZT \, dT$. The $Pb_{0.99875}(GaSb)_{0.00125}Zn_{0.01}Se_{1.01}$-0.3%Cu sample exhibited an excellent $ZT_{avg}$ of ~1.01 in the temperature range of 300–873 K. A comparison of the $PF_{avg}$ and $ZT_{avg}$ values with those of previously reported PbSe-based materials is shown in Fig. 8a, b, respectively. The $Pb_{0.99875}(GaSb)_{0.00125}Zn_{0.01}Se_{1.01}$-0.3% Cu sample exhibited the highest $PF_{avg}$ and $ZT_{avg}$ values among all the Te-free n-type PbSe-based thermoelectric materials.

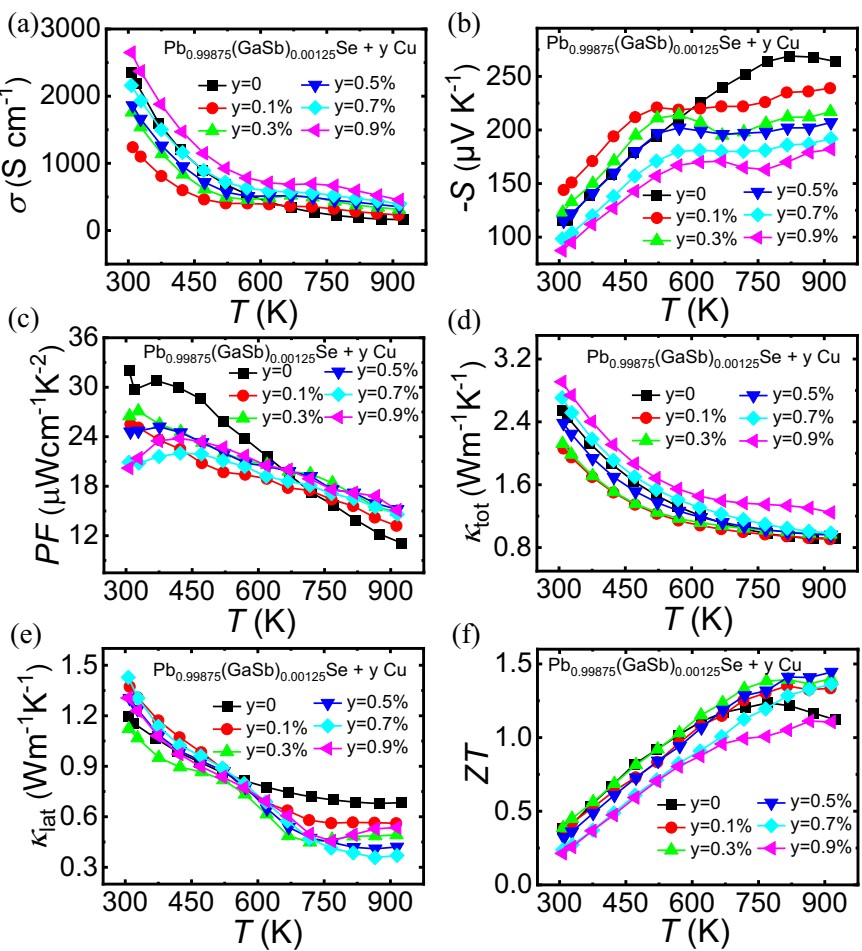

**Fig. 6 | Thermoelectric properties as a function of temperature for the $Pb_{0.99875}(GaSb)_{0.00125}Se$-yCu ($y = 0$, 0.1%, 0.3%, 0.5%, 0.7%, and 0.9%) samples.** a Electrical conductivity, $\sigma$; b Seebeck coefficient, $S$; c power factor, $PF$; d total ($\kappa_{tot}$) and e lattice ($\kappa_{lat}$) thermal conductivities; and f figure of merit, $ZT$.

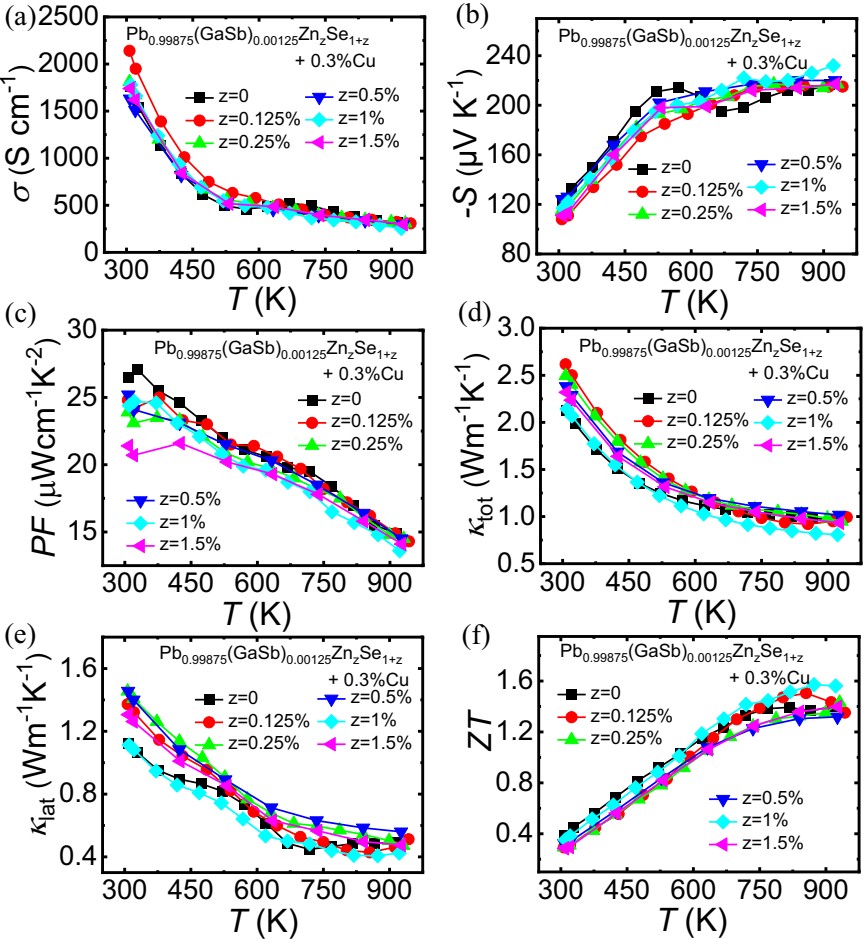

**Fig. 7 | Thermoelectric properties as a function of temperature for the $Pb_{0.99875}(GaSb)_{0.00125}Zn_zSe_{1+z}$-0.3%Cu ($z = 0$, 0.125%, 0.25%, 0.5%, 1%, and 1.5%) samples.** **a** Electrical conductivity, $\sigma$; **b** Seebeck coefficient, $S$; **c** power factor, $PF$; **d** total ($\kappa_{tot}$) and **e** lattice ($\kappa_{lat}$) thermal conductivities; and **f** figure of merit, $ZT$.

## Discussion

We rationally developed n-type PbSe solid solutions with excellent thermoelectric performance via GaSb semiconductor doping. DFT calculations and microstructural analysis showed that the Ga and Sb atoms co-occupied the Pb sites, causing local structural distortion. The DFT calculations further revealed that GaSb doping significantly enhanced the Seebeck coefficient by flattening the two conduction bands. Moreover, GaSb doping reduced the contribution of the 6p orbitals of Pb to the $L$ conduction band, apparently resulting in its upward shift and promotion of near energy convergence of the two conduction bands. Consequently, the $Pb_{0.99875}(GaSb)_{0.00125}Se$ sample exhibited a high $PF$ value of ~32 µW cm$^{-1}$ K$^{-2}$ at 300 K. Moreover, the $\kappa_{lat}$ value of the doped $Pb_{0.99875}(GaSb)_{0.00125}Zn_{0.01}Se_{1.01}$-0.3% Cu sample decreased to ~0.4 W m$^{-1}$ K$^{-1}$ at 873 K owing to the added local structural distortions of interstitial Cu atoms and discordant Zn atoms. Consequently, a record-high $ZT_{avg}$ value of ~1.01 was obtained in the temperature range of 300–873 K for a completely Te-free PbSe-based material. This study demonstrates that forming a stable solid solution and inducing local structural distortions via compound doping can favorably modify the electronic strucutre to significantly improve thermoelectric properties, thereby providing valuable insights for other thermoelectric material systems.

## Methods
### Synthesis

High-purity materials were used as received. Pb wire (99.99%) and Ga shots (99.999%) were purchased from Beijing Hawk Science & Technology Co., Ltd (China). Sb shots (99.99%), Se shots (99.99%), Cu shots

(99.99%), and Zn block (99.99%) were obtained from Hebei Luohong Technology Co., Ltd (China). A GaSb semiconductor was synthesized using a stoichiometric ratio of the Ga and Sb shots. The tubes containing the raw materials were flame-sealed at a pressure of ~$2 \times 10^{-3}$ torr, heated to 1173 K for 10 h, and then soaked at this temperature for 5 h. Finally, the samples were cooled to room temperature for 12 h. The synthesized GaSb ingots will act as dopants. The PXRD pattern of the GaSb sample shown in Fig. S17 confirms that the sample is the pure phase. At the same time, high-resolution X-ray photoelectron spectroscopy (XPS) analysis, as depicted in Fig. S18, shows that the Ga element is positive trivalent, and the Sb element is positive pentavalent. All samples with a predetermined nominal stoichiometric ratio: $Pb_{1-x}(GaSb)_xSe$ ($x = 0$, 0.05%, 0.075%, 0.1%, 0.125%, 0.15%, and 0.175%), $Pb_{0.99875}(GaSb)_{0.00125}Se$-yCu ($y = 0$, 0.1%, 0.3%, 0.5%, 0.7%, and 0.9%), and $Pb_{0.99875}(GaSb)_{0.00125}Zn_zSe_{1+z}$-0.3%Cu ($z = 0$, 0.125%, 0.25%, 0.5%, 1%, and 1.5%) were synthesized via a facile melting-quenching process. The tubes containing the raw materials were flame-sealed at a pressure of ~$2 \times 10^{-3}$ torr, heated to 1473 K for 12 h, and then soaked at this temperature for 6 h. Finally, the samples were quenched in ice water to room temperature. For a typical sample, the following amounts of the raw materials were used: Pb (8 g, 38.61 mmol), GaSb (0.0093 g, 0.04857 mmol), Se (3.083 g, 39.04 mmol), Cu (0.0074 g, 0.11645 mmol), and Zn (0.0253 g, 0.3868 mmol) for preparing ~11 g ingot sample of $Pb_{0.99875}(GaSb)_{0.00125}Zn_{0.01}Se_{1.01}$-0.3% Cu.

### Densification

The ingots were ground into a fine powder using a mortar and pestle, loaded into a 12.7 mm graphite die, and sintered using the spark

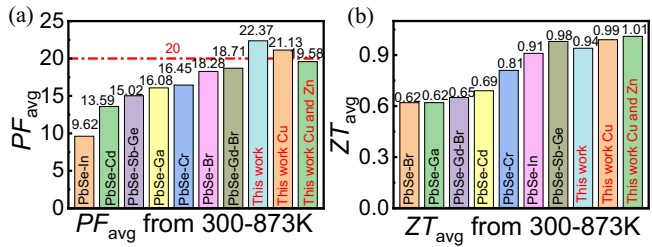

**Fig. 8 | $PF_{avg}$ and $ZT_{avg}$ of n-type Te-free PbSe-based thermoelectric materials.** Comparison of the $PF_{avg}$ (**a**) and $ZT_{avg}$ **b** values in the temperature range of 300–873 K with those of the n-type Te-free PbSe-based thermoelectric materials[37,38,49–51,56,57].

plasma sintering (SPS) technique (LABOX-110, Sinter Land Inc., Japan) at 923 K for 5 min under a constant axial pressure of 40 MPa. Pellets with relative densities greater than 96% were obtained (Tables S1, S2, and S3).

### Powder X-ray diffraction (PXRD) characterization
The PXRD patterns of the powder samples were obtained for purity analysis using a Miniflex powder X-ray diffractometer (Rigaku Corporation, Japan) with Ni-filtered Cu $K_\alpha$ ($\lambda = 1.5418$ Å) radiation operating at 40 kV and 15 mA.

### Electrical transport properties
The electrical conductivity and Seebeck coefficient of the SPS-ed samples were simultaneously measured using a CTA-3S system (Beijing Cryoall Science and Technology Co., Ltd, China). Bar-shaped samples (~10 mm × 3 mm × 3 mm) were cut from the SPS-ed pellets, coated with boron nitride, and measured in the temperature range of 300–923 K under a helium atmosphere. A thin layer of graphite paper was added to increase contact between the top and bottom of the test electrodes.

### Hall coefficient measurements
The Hall coefficient ($R_H$) was measured using a Hall effect testing system (NYMS) in a helium atmosphere. Square samples (~9 × 9 × 1 mm$^3$) were cut from the disks and polished. $R_H$ was measured using the van der Pauw technique under a magnetic field of 1.5 T. The carrier concentration $n_H$ was estimated from the $R_H$ value using the relationship $n_H = 1/(e \mid R_H \mid)$, where $e$ is the electron charge.

### Thermal conductivity
The thermal diffusivity $D$ was measured using the laser flash diffusivity method (LFA 467 MicroFlash, NETZSCH, Germany). Square pellets with dimensions of ~10 × 10 × 2 mm$^3$ coated with a thin layer of graphite were used for the measurements under continuous nitrogen flow from 300 to 923 K. $\kappa_{tot}$ was calculated using the relationship: $\kappa_{tot} = DC_p\rho$, where $C_p$ and $\rho$ are the specific heat capacity and density, respectively. $C_P$ was calculated according to the Dulong–Petit law[58]. $\rho$ was calculated using the dimensions and mass of the sample.

### Scanning/transmission electron microscopy (S/TEM) analysis
High-resolution S/TEM (HR-S/TEM) imaging was conducted using an aberration-corrected JEM-ARM200F microscope (JEOL Ltd, Japan) operating at 200 kV. A condenser aperture was selected to provide a convergence semiangle of 27.5 mrad. To prepare the electron-beam transmitted TEM specimens, the bulk samples were first ground and polished to a thickness of ~30 μm, followed by Ar ion milling (3 kV for ~30 min until a hole was formed, and ion cleaning at 0.3 kV for 40 min at a low temperature (liquid nitrogen stage). Geometric phase analysis was performed using the Strain++ program[59].

### Multislice simulation
The supercell structure files were built using a custom Python script. The supercell thickness was ~5 nm. HR-S/TEM simulations were performed using the abTEM[60] Python package at 200 kV, a defocus of 2.5 nm, slice thickness of 2 Å, and convergence angle of 27.5 mrad. Further details of the simulation can be found elsewhere[61].

### Electronic and crystal structures calculations
Density functional theory (DFT)-based first-principles calculations were conducted using the projector augmented-wave method (PAW) method implemented in the Vienna Ab-initio Simulation Package (VASP)[62]. The exchange-correlation energy was determined using the Perdew-Burke-Ernzerhof (PBE) formulation of the generalized gradient approximation (GGA)[63]. The energetic cut-off of 500 eV and total energy convergence of below $10^{-5}$ eV were used for plane-wave basis sets. The $3 \times 3 \times 3$ supercell constructed from the rock-salt primitive cell of PbSe was employed to assess the electronic structure of the pristine $Pb_{27}Se_{27}$. It is important to note that for GaSb-substituted PbSe, we first evaluated multiple possible configurations and performed band structure calculations for the energetically most favorable configuration. The different configurations of GaSb-substituted PbSe ($GaSbPb_{26}Se_{27}$) without structural relaxation are shown in Fig. S19a–e. Specifically, Fig. S1a (configuration 1) illustrates the configuration where Ga and Sb are substituted for one Pb atom in PbSe. Fig. S19b (configuration 2) and Fig. S19c (configuration 3) depict configurations where Ga substitutes one Pb atom while the Sb atom is positioned in an interstitial site close to (Fig. S19b) or far from (Fig. S19c) the Ga atom. Additionally, Fig. S19d (configuration 4) and S19e (configuration 5) show configurations where Sb substitutes one Pb atom while the Ga atom is located in an interstitial site close to (Fig. S19d) or far from (Fig. S19e) the Sb atom. Fig. S19f–j are the corresponding configurations of Fig. S19a–e, with all ionic positions relaxed to equilibrium until the calculated Hellmann-Feynman forces on each atom were less than 0.02 eV/Å. After relaxation, we observed that Fig. S19a, S19b, and S19d exhibit similar Ga and Sb substitution configurations at one Pb site, with a total ground-state energy of −223.48 eV. In contrast, the total ground-state energies for the configurations in Fig. S19c and S19e are −222.65 and −222.61 eV, respectively. The energy differences indicate a preference for Ga and Sb to be located at one Pb site. Consequently, the electronic structures of the GaSb-substituted PbSe ($Pb_{26}(GaSb)_1Se_{27}$) were calculated based on the configuration shown in Fig. S19a. Moreover, the band structure of the $5 \times 5 \times 5$ supercell of $Pb_{124}GaSbSe_{125}$ is calculated for closing the experimental value.

### Reporting summary
Further information on research design is available in the Nature Portfolio Reporting Summary linked to this article.

## Data availability
All data supporting this study and its findings are available within the article and its Supplementary Information.

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

## Acknowledgements

This study was supported in part by the National Natural Science Foundation of China (52472191 and 52102218, awarded to Z.-Z.L.), the National Key Research and Development Program of China (2020YFA0710303, awarded to Y.Y.), and the Fujian Science & Technology Innovation Laboratory for Optoelectronic Information of China (2021ZZ127 and 2023RC103, awarded to Z.-Z.L.). The authors gratefully acknowledge the Minjiang Scholar Professorship (GXRC-21004, awarded to Z.-Z.L.), the State Key Laboratory of Structure Chemistry (20240010, awarded to Z.-Z.L.), and the Natural Science Foundation of Fujian Province of China (2021J01594, awarded to Z.-Z.L.). This study was supported by the Ministry of Education (MOE) Academic Research Fund (AcRF) Tier 1 (RG128/21 and RT6/22, awarded to Q.Y.) and MOE Tier 2 (MOE-T2EP50223-0003, awarded to Q. Y.). This study was conducted at the Northwestern University and was supported by a grant from the U.S. Department of Energy, Office of Science, and Office of Basic Energy Sciences under award number DE-SC0024256, awarded to V.P.D. and M.G.K. (synthesis and HRTEM, thermal/charge ttransport of TE semiconductors).

## Author contributions

Z.-Z.L., Q.Y., and M.G.K. conceived the idea and supervised the overall experiments. J.Z., H.-H.C., and Y.L. designed the related experiments. J.Z. and H.-H.C. prepared materials and measured the thermoelectric properties. Y.L. characterized the microstructure. H.M. carried out the DFT calculations. J.Z., H.-H.C., Y.L., Y.Y., V.P.D., Z.Z., Z.-Z.L., Q.Y., and M.G.K. discussed the results. J.Z., H.-H.C., and Y.L. wrote the paper with the help of all the authors.

## Competing interests

The authors declare no competing interests.

## Additional information

[1]Key Laboratory of Advanced Materials Technologies, International (HongKong Macao and Taiwan) Joint Laboratory on Advanced Materials Technologies, College of Materials Science and Engineering, Fuzhou University, Fuzhou, China. [2]Fujian Science & Technology Innovation Laboratory for Optoelectronic Information of China, Fuzhou, Fujian, China. [3]Mechanical and Electrical Engineering Practice Center, Fuzhou University, Fuzhou, China. [4]Department of Materials Science and Engineering, Northwestern University, Evanston, IL, USA. [5]State Key Laboratory of Photocatalysis on Energy and Environment, Fuzhou University, Fuzhou, China. [6]School of Materials Science and Engineering, Nanyang Technological University, Singapore, Singapore. [7]Eco-materials and Renewable Energy Research Center, College of Engineering and Applied Sciences, Nanjing University, Nanjing, China. [8]National Laboratory of Solid State Microstructures, Nanjing University, Nanjing, China. [9]Department of Chemistry, Northwestern University, Evanston, IL, USA. [10]These authors contributed equally: Jing Zhou, Hong-Hua Cui, Yukun Liu. ✉e-mail: zzluo@fzu.edu.cn; alexyan@ntu.edu.sg; m-kanatzidis@northwestern.edu

