## [Transparent Peer Review file · Nature Communications]

Conduction band convergence and local structure distortion for superior thermoelectric performance of GaSb-doped n-type PbSe thermoelectrics

Corresponding Author: Professor Mercuri Kanatzidis

Version 0:

Reviewer comments:

Reviewer #1

(Remarks to the Author)

The work done by Zhou et al. lacks sufficient explanations and requires clarification in both experimental data and theory. It requires major revisions before it can be considered for publication. The authors should address the following comments.

1. The study claims that conduction band convergence occurs through GaSb doping in PbSe. However, this claim lacks sufficient support. Experimental data, particularly the Ioffe-Pisarenko plot, does not show the expected increase in effective mass. Additionally, the authors' theoretical calculations were performed for around 4% GaSb doping (which is not consistent with the experimental conditions).
2. The authors should clarify whether they used GaSb as a compound for doping in PbSe or separate Ga and Sb elements. If a compound was used, how can they confirm that it dissociated into separate Ga and Sb atoms and then co-doped onto the Pb sites? If separate elements were used, why did they use twice the number of Pb vacancies (as they did experimentally with both Ga and Sb)? This point needs further clarification.
3. Most of the data presented shows minimal variation and lies within the error limits. The authors should include error bars for all data to provide a clearer representation of the results.
4. The abstract states that doping with GaSb flattens the conduction band and reduces the energy difference (ΔE_V) between the Σ and L conduction bands, thereby improving the Seebeck coefficient. However, the results in the paper show that the Seebeck coefficient decreases monotonically with GaSb doping. This discrepancy should be addressed.
5. For better comparison, the authors should include data for pristine PbSe alongside the doped samples.
6. In the introduction, the authors mention that the large energy difference (ΔE_V) between the first (L) and second (Σ) conduction bands in p-type PbSe poses a challenge for achieving conduction band convergence and a high PF. However, references 40 and 41 do not support this statement and should be revisited.
7. In the introduction, the authors hypothesize that introducing compounds with high hole mobility (μ_H) into the PbSe matrix could result in a high μ_H in the solid solution without forming a secondary phase. However, carrier mobility in compounds depends on interatomic forces within the unit cell and defect scattering. It's not guaranteed that doping with a compound of high mobility will increase the mobility of the host matrix.
8. The authors state that GaSb doping in PbSe leverages the unique co-occupancy of Ga and Sb atoms on Pb sites in the PbSe matrix to create n-type character. However, Ga is electropositive, and Sb is electronegative compared to Pb. The authors should clarify this point.
9. The authors should correct the text regarding the use of ZnSe to reduce thermal conductivity experimentally, in the introduction they mentioned Zn atoms.
10. The XRD data indicates that the lattice parameters increase with GaSb doping. This is unexpected because Ga and Sb are both smaller than Pb, and it is unlikely that both Ga and Sb atoms occupy the Pb sites simultaneously. The authors should explain this discrepancy.
11. The authors state that the dynamic doping effect of Cu ions generates extra charge carriers, as Cu ions go into interstitial sites. However, the electrical conductivity decreases with Cu addition in most samples. This contradiction needs to be addressed.
12. The authors should provide a more detailed description of each step in their synthesis method for clarity and reproducibility.
13. The authors should clarify the superlattice structure of GaSb-doped PbSe used for the band structure calculations.

Reviewer #2

(Remarks to the Author)

In this paper, the authors report that the thermoelectric performance of n-PbSe was improved by doping GaSb compounds due to the effects of band convergence and lattice distortion.

Band convergence and lattice distortion are now well-known strategies to improve thermoelectric performance in thermoelectric materials research, and many similar studies have been performed on similar PbTe materials up to less than 10 years ago. Since then, the same focus has shifted from PbTe to similar SnTe and PbSe. In particular, several papers on performance enhancement of PbSe have been published very recently (2023-2024) (see, for example, R1-R5), where values equivalent to the highest achieved thermoelectric performance in this paper have already been reported. Therefore, this paper does not have much advantage in thermoelectric performance improvement.

However, this paper might have a novel feature regarding the crystal structure of the samples, apart from other papers. The authors claim that two atoms, one Ga and one Sb, are substituted for one Pb deficiency. If this is true, it is very exciting regarding materials science research. The authors are trying to prove the substitution by microstructure observation and electronic structure calculation. However, I could not understand the proof. More details below. In the end, I do not recommend the publication of this paper.

1. Band structure calculation

The band structure of PbSe in Fig. 6 is quite different from those in other papers (R6, R7), which were calculated by the same VASP program. I understand that the band structure depends somewhat on the calculation program. However, in the study of band convergence using band edge structures, the shapes of the band structures seem to have a large influence.

The method of determining the atomic configuration used in the band calculations, as described in lines 439-458 of the main text and Fig. S1, is acceptable. However, there is no discussion of the stability of Pb₂₆GaSbSe₂₇. For example, its stability could be discussed by the formation energies of Pb₂₇Se₂₇, Pb₂₆GaSbSe₂₇, GaSb, and Pb.

The chemical composition of the sample is, for example, Pb_{0.99875}(GaSb)_{0.00125}Se, whereas the chemical composition of the band calculation target is Pb₂₆(GaSb)Se₂₇. The GaSb content differs by a factor of about 30 between them. Is this band calculation result valid for the discussion of sample properties?

2. Microstructural analysis

I could not understand from Fig. 3 alone that GaSb is replacing Pb deficiency, although I understand that there is lattice distortion.

The interior of the sample appears to be locally Ga/Sb doped, rather than GaSb doped throughout. In other words, it appears to be a two-phase or composite structure, where the parent phase of the first phase is PbSe and the second phase is Ga/Sb-doped PbSe with a distorted lattice.

If so, the presence of the second phase may have the effect of reducing thermal conductivity. However, the thermoelectric properties would be roughly the volume average of the thermoelectric properties of each phase, would they not? However, since the volume fraction of phase 2 is small, it may not have much effect.

R1) B. Ge et al., *Energy Environ. Sci.* 16(2023)3994, "Engineering an atomic-level crystal lattice and electronic band structure for an extraordinarily high average thermoelectric figure of merit in n-type PbSe":

The abstract says "the best composition achieves a remarkably high average power factor of 24 $\mu\text{W cm}^{-1}\text{K}^{-2}$ from 300 to 823 K, with a substantially depressed lattice thermal conductivity of 0.2 $\text{W m}^{-1}\text{K}^{-1}$ at 723 K. With a ZT of 0.55 at 300 K, an average ZT is 1.30 from 400 to 823 K".

R2) J. Cai et al., *Adv. Funct. Mater.* 34(2024)2311217, "Giant Band Convergence and High Thermoelectric Performance in n-Type PbSe Induced by Spin-Orbit Coupling":

The abstract says "Owing to the achieved giant band convergence and the low thermal conductivity, a high peak zT of 1.75 at 850 K and an outstanding average zT of 1.04 are obtained."

R3) D. Zhang et al., *Scripta Materialia*, 244(2024)116003, "High thermoelectric performance of PbSe via a synergistic band engineering and dislocation approach":

The abstract says "a high average ZT_{ave} 0.97 between 300 and 873 K is obtained for Pb_{0.9985}Gd_{0.0015}Se-0.2 %Cu."

R4) S. Wang et al., *Energy Environ. Sci.* 17(2024)2588, "Realizing high-performance thermoelectric modules through enhancing the power factor via optimizing the carrier mobility in n-type PbSe crystals":

The abstract says "the ZT values of the Pb_{1.006}Se+0.0016Al crystal reached 0.5 at 300 K, 1.5 at 673 K, and the average ZT (ZT_{ave}) reached 1.1 at 300-773 K."

R5) S. Wu et al., *J. Mater. Chem. A* 12(2024)26013, "Optimizing the thermoelectric performance of n-type PbSe through dynamic doping driven by entropy engineering":

The abstract says "the thermoelectric properties of $\text{Pb}_{0.875}\text{Sn}_{0.125}\text{Se}_{0.5}\text{Te}_{0.25}\text{S}_{0.25-2\text{at}\%}\text{Cu}$ are remarkably improved, with a maximum dimensionless merit ZT_{max} of 1.46 at 623 K and an average dimensionless merit ZT_{ave} of 1.15 at 300-700 K."

R6) K. Hummer et al., Phys. Rev. B 75(2007)195211, "Structural and electronic properties of lead chalcogenides from first principles":

In Fig.1, an energy difference is 3 eV between L and Gamma points of CBM for PbSe, which was calculated by VASP.

R7) Y. Zhang et al., Phys. Rev. B 80(2009)024304, "Thermodynamic properties of PbTe, PbSe, and PbS: First-principles study":

In Fig.1(b), an energy difference is 0.9 eV between L point and Sigma axis of CBM for PbSe, which was calculated by VASP.

Reviewer #3

(Remarks to the Author)

Article Title: "Conduction band convergence and local structure distortion for a superior thermoelectric performance of GaSb-doped n type PbSe thermoelectrics" The article reports the synthesis of the n-type PbSe solid solutions via GaSb doping resulting in superior TE properties. DFT calculations were also carried out to prove the high Seebeck coefficient because of band convergence. Thermal conductivity was further reduced by addition of interstitial Cu atoms and Zn alloying as a result of which ZT values were enhanced upto 1.57. Results are interesting. Overall the manuscript have good introduction, experimental details are adequate. Discussion of results is very good. I have following questions/suggestions for revisions.

1. As written in the manuscript $\text{Pb}_{1-x}(\text{GaSb})_x\text{Se}$ ($x = 0, 0.05\%, 111\ 0.075\%, 0.1\%, 0.125\%, 0.15\%, \text{ and } 0.175\%$). Non-doped sample data is only provided in the XRD graphs. Authors are suggested to add the non-doped sample data for comparison in all graphs to account for the actual increment in TE properties with this doping.
2. Figure 6 shows band structure and density of states. Indicate the fermi level position in both graphs. If it is at zero then the explanation of bands crossing fermi level should also be added.
3. Can authors elaborate more on the behavior of klatt upon addition of interstitial Cu atoms. Why the decrease was dominant in only above 573K.

Reviewer #4

(Remarks to the Author)

Zhou, Jing. et al. demonstrated high thermoelectric figure of merit zT in PbSe by advanced strategies, including conduction band convergence and local structure distortion. High average zT (~ 1.01 in 300 to 873 K) was obtained in $\text{Pb}_{0.99875}(\text{GaSb})_{0.00125}\text{Se}_{0.3\%}\text{Cu}_{-1\%}\text{ZnSe}$. The conclusions and findings in this study are strongly supported by experimental (XRD, SEM and TEM observation, electrical and transport properties measurements) and theoretical (DFT calculation of electronic structures) results. Therefore, this work should be of widespread interest to researchers in the fields of thermoelectrics and natural sciences. There is no major concern. I would recommend it for acceptance in its current form.

Version 1:

Reviewer comments:

Reviewer #2

(Remarks to the Author)

In the revised manuscript, the authors provide a more detailed microstructural analysis of the samples to prove that GaSb substitutes for the Pb site. This makes it more convincing. On the other hand, they do not discuss the stability of the samples using electronic structure calculation on the total energy, which I pointed out in the first review. That is, it is not clarified which is larger, the formation energy of $\text{Pb}_{26}(\text{GaSb})\text{Se}_{27}$ or the sum of the formation energies of 26PbSe, GaSb, and Se. Also, the authors explain the superiority of this paper over the previously published ones regarding thermoelectric performance improvement. The improvements seem to be small, but they exist. I recommend the publication of this paper because this substitution is a new strategy to improve thermoelectric performance.

Reviewer #3

(Remarks to the Author)

The authors have revised the manuscript according to comments. Manuscript is now significantly improved and now I recommend it for publication

Response Letter to Reviewers

Reviewer #1 (Remarks to the Author):

The work done by Zhou et al. lacks sufficient explanations and requires clarification in both experimental data and theory. It requires major revisions before it can be considered for publication. The authors should address the following comments.

Authors' response: We are grateful for your review of the manuscript. We have carefully and significantly revised the manuscript.

1. The study claims that conduction band convergence occurs through GaSb doping in PbSe. However, this claim lacks sufficient support. Experimental data, particularly the Ioffe-Pisarenko plot, does not show the expected increase in effective mass. Additionally, the authors' theoretical calculations were performed for around 4% GaSb doping (which is not consistent with the experimental conditions).

Authors' response: We redrew the Ioffe-Pisarenko plot (Figure 5c) to more clearly show the significantly increased effective mass. The redrawn Figure 5(c) is shown as follows.

Figure 5. Thermoelectric properties as a function of temperature for the $\text{Pb}_{1-x}(\text{GaSb})_x\text{Se}$ ($x = 0, 0.05\%, 0.075\%, 0.1\%, 0.125\%, 0.15\%,$ and 0.175%) samples: (a) Electrical conductivity, σ ; (b) Seebeck coefficient, S ; and (c) Seebeck coefficient as a function of n_H at room temperature. (d) Seebeck coefficient as a function of n_H at different temperatures (300, 373, 473, 573, 673, and 773 K) for the $\text{Pb}_{0.99875}(\text{GaSb})_{0.00125}\text{Se}$ sample.

We have sorted out the specific density of states effective mass values in Table R1. The results show that the density of states effective mass of the GaSb-doped samples is larger than that of the pure PbSe.

Table R1. The density of states effective mass of the $\text{Pb}_{1-x}(\text{GaSb})_x\text{Se}$ ($x = 0, 0.05\%, 0.075\%, 0.1\%, 0.125\%, 0.15\%,$ and 0.175%) samples.

Sample	density of states effective mass
x=0	$0.43m_e$
x=0.05%	$0.48m_e$
x=0.075%	$0.52m_e$
x=0.1%	$0.52m_e$
x=0.125%	$0.59m_e$
x=0.15%	$0.57m_e$
x=0.175%	$0.47m_e$

Due to the low doping amount of GaSb, it is a challenge to expand the cell to $\text{Pb}_{799}(\text{GaSb})_1\text{Se}_{800}$ for theoretical calculations. Therefore, we calculate the band structure of the sample, which is close to the experimental value. Figure S10 shows the band structure of $\text{Pb}_{124}(\text{GaSb})_1\text{Se}_{125}$. GaSb-doping decreased the ΔE_c between the Σ and L bands from 0.33 eV for the intrinsic PbSe to 0.13 eV for the GaSb-doped PbSe. Figure 6c shows the band structure of $\text{Pb}_{26}(\text{GaSb})_1\text{Se}_{27}$. GaSb-doping decreased the ΔE_c between the Σ and L bands from 0.33 eV for the intrinsic PbSe to 0.08 eV for the GaSb-doped sample. According to the theoretical calculation results, the ΔE_c value is reduced with the GaSb content. Previous studies on p-type PbTe^1 and PbSe^2 have established precedent for using this type of evidence to support band convergence, particularly

involving the two valence bands. In our work, we demonstrate this phenomenon in GaSb-doped n-type PbSe, where achieving conduction band convergence has been more challenging due to the larger ΔE_c between the two conduction bands. Therefore, GaSb-doping can realize the conduction band convergence.

Figure S10. Electronic band structures for GaSb-doped PbSe ($\text{Pb}_{124}\text{GaSbSe}_{125}$).

Figure 6. (a, c) Electronic band structures for the intrinsic and GaSb-doped PbSe-based material, respectively. (b, d) Projected DOS for the pure and GaSb-doped PbSe-based material, respectively.

- r1. Jood, P. et al. Na Doping in PbTe: Solubility, Band Convergence, Phase Boundary Mapping, and Thermoelectric Properties. *J. Am. Chem. Soc.* **142**, 15464-15475 (2020).
- r2. Zhu, Y. et al. Multiple valence bands convergence and strong phonon scattering

lead to high thermoelectric performance in p-type PbSe. *Nat. Commun.* **13**, 4179 (2022).

2. The authors should clarify whether they used GaSb as a compound for doping in PbSe or separate Ga and Sb elements. If a compound was used, how can they confirm that it dissociated into separate Ga and Sb atoms and then co-doped onto the Pb sites? If separate elements were used, why did they use twice the number of Pb vacancies (as they did experimentally with both Ga and Sb)? This point needs further clarification.

Authors' response: The GaSb compound was used for doping. It was proved that GaSb was co-doped in the Pb site by first principles calculation and microstructure analysis.

First of all, through the first-principles calculations, multiple possible configurations were evaluated. The different configurations of GaSb-substituted PbSe ($\text{GaSbPb}_{26}\text{Se}_{27}$) without structural relaxation are shown in Figures S1(a-e). Specifically, Figure S1(a) illustrates that Ga and Sb are substituted for one Pb atom in PbSe (configuration 1). Figures S1(b) and S1(c) depict that Ga substitutes one Pb atom while the Sb atom is positioned in an interstitial site close to (Figure S1(b), configuration 2) or far from (Figure S1(c), configuration 3) the Ga atom. Additionally, Figures S1(d) and S1(e) show that Sb substitutes one Pb atom while the Ga atom is located in an interstitial site close to (Figure S1(d), configuration 4) or far from (Figure S1(e), configuration 5) the Sb atom. Figures S1(f-j) are the corresponding configurations of Figures S1(a-e), with all ionic positions relaxed to equilibrium until the calculated Hellmann-Feynman forces on each atom were less than $0.02 \text{ eV}/\text{\AA}$. After relaxation, we observed that configuration 1, configuration 2, and configuration 4 exhibit similar Ga and Sb substitution configurations at one Pb site, with a total ground-state energy of -223.48 eV . In contrast, the total ground-state energies for configuration 3 and configuration 5 are -222.65 and -222.61 eV , respectively. The energy differences indicate a preference for Ga and Sb to be located at one Pb site, as in configuration 1.

Figure S1. The possible configurations for GaSb-doped PbSe: (a) configuration 1, Ga and Sb substitute for one Pb atom; configuration 2 and configuration 3, Ga substitutes one Pb atom while the Sb atom is positioned in an interstitial site close to (b) or far from (c) the Ga atom; configuration 4 and configuration 5, Sb substitutes one Pb atom while the Ga atom is located in an interstitial site close to (d) or far from (e) the Sb atom and (f-j) are the corresponding optimize configurations of (a-e).

Secondly, as shown in Figure 1(b), the lattice parameters of the GaSb-doped PbSe increased with increasing GaSb amount. Since the atomic radii of Ga and Sb are smaller than Pb, this can be attributed to the replacement of Pb with Ga and Sb.

Figure 1. (a) XRD patterns and (b) refined lattice parameters of the $\text{Pb}_{1-x}(\text{GaSb})_x\text{Se}$ ($x = 0, 0.05\%, 0.075\%, 0.1\%, 0.125\%, 0.15\%,$ and 0.175%) samples as a function of the GaSb content.

In addition, by analyzing the atomic-resolution HAADF images, we found that the intensity line profiles from the experimental results showed additional atoms corresponding to Ga ($Z = 31$) and Sb ($Z = 51$) near the Pb sites, which exhibited considerably low intensities than Pb in the HAADF image (Figure 3e). This local structural distortion could be caused by the fact that the Ga and Sb atoms co-occupy the vacant Pb sites, as predicted by the DFT calculations.

Figure 3. S/TEM analysis of the microstructure of the $\text{Pb}_{0.99875}(\text{GaSb})_{0.00125}\text{Se}$ sample. (a) HAADF image of a representative region with the corresponding EDS maps, where all elements are homogeneously distributed. Weak signals of Ga and Sb are observed owing to a low doping concentration. (b) SAED pattern along the $[110]$ zone axis acquired from the region shown in (a). (c) Atomic-resolution HAADF image showing the local structural distortion. (d) Multislice HAADF image simulation of pure PbSe without structural distortion. (e) Intensity line profiles of the regions highlighted in (c) and (d). (f) and (g) HAADF images of representative local structural distortions in the microstructure. (h) HAADF image of an edge dislocation induced by local structural distortion. The strain distribution is shown in the inset image: $y = [2\bar{2}0]$ and $x = [00\bar{2}]$.

Additional simulation-based analyses were incorporated to better understand the configuration of Ga and Sb atoms co-occupying the vacant Pb sites. In Figure 3(c), the atomic-resolution HAADF image reveals lattice distortions resulting from GaSb co-occupying the Pb site. This observation is based on two key features: atomic radii and

contrast. In pristine PbSe, when viewed along the [110] zone axis, the cation and anion sites separate into distinct alternating atomic columns, leading to alternating bright (Pb columns, $Z = 82$) and dark contrast (Se columns, $Z = 34$). Due to the ordered occupation, the radii of the bright and dark columns remain uniform and do not vary spatially. These phenomena are corroborated by multislice simulations performed on pristine PbSe, as shown in Figure 3(d) and Figure S3(b).

When the Ga and Sb atoms co-occupied the vacant Pb sites, two representative and simplified configurations were considered to analyze how co-doping altered the HAADF image features. In the first case, for GaSb co-doping, Ga occupies the deficient Pb site, whereas Sb cannot fit within the site and shifts by $1/3$ of the unit cell along the [010] direction. The corresponding atomic model is presented as Atomic Model 1 in Figure S5. To emphasize the contrast changes, we assume this co-occupation occurs consistently along the z-direction. The multislice simulation suggests that the local lattice distortion closely resembles the experimental observations, as shown in Figure S3(a) and Figure S4(a), with both highlighted by a pair of red arrows. The line profiles indicate a drop in intensity at the cation sites for co-doping. This reduction is due to the significantly lower atomic number of Ga ($Z = 31$) compared to Pb ($Z = 82$). Additionally, the projected cation column radii decrease in the regions for co-doping, as the atomic radius of Ga ($r = 130$ pm) is smaller than that of Pb ($r = 180$ pm). Furthermore, in the line profile along the [010] direction, an additional atomic column overlapping with the Se columns is observed in the simulation, as shown in Figure S4(a). This is because Sb cannot fit within the Pb site alongside Ga, causing Sb columns to shift. The simulated results agree well with the experimental observations, as shown in Figure S3(a), where an additional atomic column overlaps with the Se column.

In the second case, Sb occupies the vacant Pb site, while Ga shifts by $1/3$ of the unit cell along the [010] direction. The corresponding atomic model is shown as Atomic Model 2 in Figure S5. We assume that this co-occupation extends uniformly along the z-direction. The multislice simulation suggests that the local lattice distortion is consistent with experimental observations, as illustrated in Figure S3(a) and Figure S4(b), with both highlighted by a pair of red arrows. The line profiles again indicate an

intensity drop at the cation sites, due to the lower atomic number of Sb ($Z = 51$) than Pb ($Z = 82$). Similar to the Atomic Model 1, the projected cation column radii decrease for co-doping, because Sb (atomic radius = 145 pm) is smaller than Pb (180 pm). Additionally, the line profile along the [010] direction reveals an extra atomic column overlapping with the Se columns, as shown in Figure S4(b). This phenomenon is because Ga cannot co-occupy the Pb site with Sb, leading to its displacement. The simulation results align well with experimental observations in Figure S3(a), where an additional atomic column overlaps with the Se column.

In summary, when GaSb co-occupies the deficient Pb site, the local image intensity of the cationic site and the radius of the atomic column decrease. Additionally, if the second atom in the GaSb pair cannot fit within the Pb site, an extra atomic column may become visible. While the precise image characteristics depend on the specific orientation of GaSb, the overall features should align with the trends discussed here. The experimentally observed local lattice distortion follows these predicted changes, further supporting the GaSb co-occupation at the deficient Pb site.

Figure S3. (a) Experimental atomic resolution HAADF image of $\text{Pb}_{0.99875}(\text{GaSb})_{0.00125}\text{Se}$ and (b) multislice simulation of pristine PbSe. The atomic model used for simulation is shown in Figure S5.

Figure S4. Multislice simulation of GaSb co-occupation with (a) Ga atom residing at the Pb site, and (b) Sb atom residing at the Pb site. The atomic models used for simulation are shown in Figure S5.

Figure S5. Atomic model for PbSe and $Pb_{1-x}(GaSb)_xSe$ with different configurations used for multislice simulation of HAADF images as shown in Figure S3 and Figure S4.

3. Most of the data presented shows minimal variation and lies within the error limits. The authors should include error bars for all data to provide a clearer representation of the results.

Authors' response: Thanks for your suggestions. In order to make the Figures clear and intuitive, we have added error bars for the sample with the highest ZT value in the

revised Supporting Information. The margin of error is 20%.

Figure S21. Figure of merit, ZT as a function of temperature for the $\text{Pb}_{0.99875}(\text{GaSb})_{0.00125}\text{Zn}_z\text{Se}_{1+z}-0.3\% \text{Cu}$ ($z = 0, 0.125\%, 0.25\%, 0.5\%, 1\%,$ and 1.5%) samples.

4. The abstract states that doping with GaSb flattens the conduction band and reduces the energy difference (ΔE_V) between the Σ and L conduction bands, thereby improving the Seebeck coefficient. However, the results in the paper show that the Seebeck coefficient decreases monotonically with GaSb doping. This discrepancy should be addressed.

Authors' response: Thanks for your comments. Seebeck coefficient can be calculated by the formula: $S = \frac{8\pi k_B^2}{3eh^2} m^* T \left(\frac{\pi}{3n_H}\right)^2$, where k_B is the Boltzmann constant, e is the electron charge, h is the Planck constant, m^* is the carrier effective mass, T is the absolute temperature, and n_H is the carrier concentration, respectively. From the formula, the Seebeck coefficient is directly proportional to m^* and inversely proportional to $n_H^{2/3}$. Although GaSb-doping increases the effective mass by flattening the conduction band (increasing S), it also significantly increases the n_H (decreasing S). With the increased effective mass, the S still decreased because of the significantly raised n_H .

Figure R1. (a) carrier concentration and (b) density of states effective mass of $\text{Pb}_{1-x}(\text{GaSb})_x\text{Se}$ ($x = 0, 0.05\%, 0.075\%, 0.1\%, 0.125\%, 0.15\%,$ and 0.175%) as a function of GaSb semiconductor content at 300 K.

5. For better comparison, the authors should include data for pristine PbSe alongside the doped samples.

Authors' response: Thanks for your suggestions. Data for the pristine PbSe have been added to the revised manuscript.

Figure 4. (a) Temperature-dependent Hall coefficient, R_H ; (b) carrier concentration, n_H ; and (c) carrier mobility, μ_H for the $\text{Pb}_{1-x}(\text{GaSb})_x\text{Se}$ ($x = 0, 0.05\%, 0.075\%, 0.1\%, 0.125\%, 0.15\%,$ and 0.175%) samples. (d) μ_H as a function of n_H for n-type PbSe-based thermoelectrics.

Figure 5. Thermoelectric properties as a function of temperature for the $\text{Pb}_{1-x}(\text{GaSb})_x\text{Se}$ ($x = 0, 0.05\%, 0.075\%, 0.1\%, 0.125\%, 0.15\%,$ and 0.175%) samples: (a) Electrical conductivity, σ ; (b) Seebeck coefficient, S ; and (c) Seebeck coefficient as a function of n_H at room temperature. (d) Seebeck coefficient as a function of n_H at different temperatures (300, 373, 473, 573, 673, and 773 K) for the $\text{Pb}_{0.99875}(\text{GaSb})_{0.00125}\text{Se}$ sample.

Figure 7. Thermal conductivity as a function of temperature for the $\text{Pb}_{1-x}(\text{GaSb})_x\text{Se}$ ($x = 0, 0.05\%, 0.075\%, 0.1\%, 0.125\%, 0.15\%,$ and 0.175%) samples: (a) total (κ_{tot}) and (b) lattice (κ_{lat}) thermal conductivities.

Figure 8. Figure of merit (ZT) as a function of temperature for the $Pb_{1-x}(GaSb)_xSe$ ($x = 0, 0.05\%, 0.075\%, 0.1\%, 0.125\%, 0.15\%$, and 0.175%) samples.

Figure S11. Temperature-dependent (a) electronic thermal conductivity, κ_{ele} ; (b) thermal diffusivity, D ; (c) heat capacity, C_p ; and (d) Lorenz numbers, L for $Pb_{1-x}(GaSb)_xSe$ samples ($x = 0, 0.05\%, 0.075\%, 0.1\%, 0.125\%, 0.15\%$, and 0.175%).

6. In the introduction, the authors mention that the large energy difference (ΔE_v) between the first (L) and second (Σ) conduction bands in n-type $PbSe_x$ poses a challenge for achieving conduction band convergence and a high PF . However, references 40 and 41 do not support this statement and should be revisited.

Authors' response: We have deleted references 40 and 41 in the revised manuscript. The new reference 43 (*J. Am. Chem. Soc.* **142**, 15172-15186, 2020) has been cited. The newly cited references are as follows: C. Zhou et al., *J. Am. Chem. Soc.* **142**, 15172-15186 (2020). "Exceptionally High Average Power Factor and Thermoelectric Figure of Merit in n-type PbSe by the Dual Incorporation of Cu and Te". The references mention: *In contrast, the band convergence is impossible in n-type PbQ due to single conduction band at the L point in the vicinity of the Fermi level. Consequentially, proven strategies to enhance its power factor has long been elusive.*

7. In the introduction, the authors hypothesize that introducing compounds with high hole mobility (μ_H) into the PbSe matrix could result in a high μ_H in the solid solution without forming a secondary phase. However, carrier mobility in compounds depends on interatomic forces within the unit cell and defect scattering. It's not guaranteed that doping with a compound of high mobility will increase the mobility of the host matrix.

Authors' response: You are correct that μ_H in compounds depends on interatomic forces within the unit cell and defect scattering. Doping with a compound with high μ_H is not guaranteed to increase the μ_H of the host matrix. In the revised manuscript, the μ_H of pure PbSe samples is lower than that of GaSb-doped samples. This is a function of lattice plainification, which has been used to improve the μ_H of PbSe-based thermoelectric materials.^{r3,r4} A small amount of GaSb-doping can achieve lattice plainification, effectively reducing the scattering effect of Pb vacancy on carriers and improving μ_H .

r3. Liu, S. et al. Lattice Plainification Leads to High Thermoelectric Performance of P-Type PbSe Crystals. *Adv. Mater.* **36**, 2401828 (2024).

r4. Qin, Y. et al. Grid-plainification enables medium-temperature PbSe thermoelectrics to cool better than Bi₂Te₃. *Science* **383**, 1204-1209 (2024).

8. The authors state that GaSb doping in PbSe leverages the unique co-occupancy of Ga and Sb atoms on Pb sites in the PbSe matrix to create n-type character. However, Ga is electropositive, and Sb is electronegative compared to Pb. The authors should clarify this point.

Authors' response: Based on the Pauling electronegativity scale, the electronegativity of Pb, Ga, and Sb are 2.33, 1.81, and 2.05, respectively. Both Ga and Sb are electropositive compared with Pb. Thus, Ga and Sb show the positive valence states.

H 2.20 (0.00)																	He 3.20
Li 0.98 0.97 (0.01)	Be 1.57 1.47 (0.10)											B 2.04 2.01 (0.03)	C 2.55 2.50 (0.05)	N 3.04 3.07 (-0.03)	O 3.44 3.50 (-0.06)	F 3.98 4.10 (-0.12)	Ne 5.10
Na 0.93 1.01 (-0.08)	Mg 1.31 1.23 (0.08)											Al 1.61 1.47 (0.14)	Si 1.90 1.74 (0.16)	P 2.19 2.06 (0.13)	S 2.58 2.44 (0.14)	Cl 3.16 2.83 (0.33)	Ar 3.30
K 0.82 0.91 (-0.09)	Ca 1.00 1.04 (-0.04)	Sc 1.36 1.20 (0.16)	Ti 1.54 1.32 (0.22)	V (II) 1.63 1.45 (0.18)	Cr (II) 1.66 1.56 (0.10)	Mn (II) 1.55 1.60 (-0.05)	Fe (II) 1.83 1.64 (0.19)	Co (II) 1.88 1.70 (0.18)	Ni (II) 1.91 1.75 (0.16)	Cu (I) 1.90 1.75 (0.15)	Zn 1.65 1.66 (-0.01)	Ga (III) 1.81 1.82 (-0.01)	Ge (IV) 2.01 2.02 (-0.01)	As (III) 2.18 2.20 (-0.02)	Se 2.55 2.48 (0.07)	Br 2.96 2.74 (0.22)	Kr 3.10
Rb 0.82 0.89 (-0.07)	Sr 0.95 0.99 (-0.04)	Y 1.22 1.11 (0.11)	Zr (II) 1.33 1.23 (0.11)	Nb 1.60 1.23 (0.37)	Mo (II) 2.16 1.30 (0.86)	Tc 1.99 1.36 (0.54)	Ru 2.20 1.42 (0.78)	Rh 2.28 1.45 (0.83)	Pd 2.20 1.35 (0.85)	Ag 1.93 1.42 (0.51)	Cd 1.69 1.48 (0.23)	In 1.78 1.49 (0.29)	Sn (IV) 1.96 1.72 (0.24)	Sb (III) 2.05 2.01 (0.23)	Te 2.10 2.01 (0.09)	I 2.66 2.21 (0.45)	Xe 2.40
Cs 0.79 0.86 (-0.07)	Ba 0.89 0.97 (-0.08)	La 1.10 1.08 (0.02)	Hf 1.30 1.23 (0.07)	Ta 1.50 1.33 (0.17)	W 2.36 1.40 (0.96)	Re 2.36 1.46 (0.44)	Os 2.20 1.52 (0.68)	Ir 2.20 1.55 (0.65)	Pt 2.28 1.44 (0.84)	Au 2.54 1.42 (1.12)	Hg 2.00 1.44 (0.56)	Tl (I) 1.82 1.44 (0.38)	Pb (IV) 2.33 1.55 (0.78)	Bi 2.02 1.67 (0.35)			Rn

Figure R2. Periodic table of electronegativity using the Pauling scale (DOI:10.19789/j.1004-9398.1990.02.006.).

High-resolution X-ray photoelectron spectroscopy (XPS) analysis, as depicted in Figure S22, shows that the Ga element is positive trivalent, and the Sb element is positive pentavalent.

Figure S22. High-resolution XPS spectra of $\text{Pb}_{0.99875}(\text{GaSb})_{0.00125}\text{Se}$ sample: (a) Ga 3d and (b) Sb 3d.

9. The authors should correct the text regarding the use of ZnSe to reduce thermal conductivity experimentally, in the introduction they mentioned Zn atoms.

Authors' response: Thanks for your suggestions. In the experiment, we separately added equal stoichiometric ratios Zn and Se, which is similar to the Zn-doping in the Pb position. In the revised manuscript, we have changed ZnSe to Zn.

10. The XRD data indicates that the lattice parameters increase with GaSb doping. This is unexpected because Ga and Sb are both smaller than Pb, and it is unlikely that both Ga and Sb atoms occupy the Pb sites simultaneously. The authors should explain this discrepancy.

Authors' response: Thanks for your comments. The increase in lattice parameters with GaSb-doping is an important support for both Ga and Sb atoms occupying the Pb sites. Beyond that, the first-principles calculations, atomic-resolution HAADF images, and TEM images simulated by the multislice method strongly support co-occupation. First of all, through the first-principles calculations, multiple possible configurations were evaluated. The different configurations of GaSb-substituted PbSe ($\text{GaSbPb}_{26}\text{Se}_{27}$) without structural relaxation are shown in Figures S1(a-e). Specifically, Figure S1(a) illustrates that Ga and Sb are substituted for one Pb atom in PbSe (configuration 1). Figures S1(b) and S1(c) depict that Ga substitutes one Pb atom while the Sb atom is positioned in an interstitial site close to (Figure S1(b), configuration 2) or far from (Figure S1(c), configuration 3) the Ga atom. Additionally, Figures S1(d) and S1(e) show that Sb substitutes one Pb atom while the Ga atom is located in an interstitial site close to (Figure S1(d), configuration 4) or far from (Figure S1(e), configuration 5) the Sb atom. Figures S1(f-j) are the corresponding configurations of Figures S1(a-e), with all ionic positions relaxed to equilibrium until the calculated Hellmann-Feynman forces on each atom were less than $0.02 \text{ eV}/\text{\AA}$. After relaxation, we observed that configuration 1, configuration 2, and configuration 4 exhibit similar Ga and Sb substitution configurations at one Pb site, with a total ground-state energy of -223.48 eV . In contrast, the total ground-state energies for configuration 3 and configuration 5 are -222.65 and -222.61 eV , respectively. The energy differences indicate a preference for Ga and Sb to be located at one Pb site, as in configuration 1.

Figure S1. The possible configurations for GaSb-doped PbSe: (a) configuration 1, Ga and Sb substitute for one Pb atom; configuration 2 and configuration 3, Ga substitutes one Pb atom while the Sb atom is positioned in an interstitial site close to (b) or far from (c) the Ga atom; configuration 4 and configuration 5, Sb substitutes one Pb atom while the Ga atom is located in an interstitial site close to (d) or far from (e) the Sb atom and (f-j) are the corresponding optimize configurations of (a-e).

Secondly, as shown in Figure 1(b), the lattice parameters of the GaSb-doped PbSe increased with increasing GaSb amount. Since the atomic radii of Ga and Sb are smaller than Pb, this can be attributed to the replacement of Pb with Ga and Sb.

Figure 1. (a) PXRD patterns and (b) refined lattice parameters of the $\text{Pb}_{1-x}(\text{GaSb})_x\text{Se}$ ($x = 0, 0.05\%, 0.075\%, 0.1\%, 0.125\%, 0.15\%,$ and 0.175%) samples as a function of the GaSb content.

In addition, by analyzing the atomic-resolution HAADF images, we found that the intensity line profiles from the experimental results showed additional atoms corresponding to Ga ($Z = 31$) and Sb ($Z = 51$) near the Pb sites, which exhibited considerably low intensities than Pb in the HAADF image (Figure 3e). This local structural distortion could be caused by the fact that the Ga and Sb atoms co-occupy the vacant Pb sites, as predicted by the DFT calculations.

Figure 3. S/TEM analysis of the microstructure of the $\text{Pb}_{0.99875}(\text{GaSb})_{0.00125}\text{Se}$ sample. (a) HAADF image of a representative region with the corresponding EDS maps, where all elements are homogeneously distributed. Weak signals of Ga and Sb are observed owing to a low doping concentration. (b) SAED pattern along the $[110]$ zone axis acquired from the region shown in (a). (c) Atomic-resolution HAADF image showing the local structural distortion. (d) Multislice HAADF image simulation of pure PbSe without structural distortion. (e) Intensity line profiles of the regions highlighted in (c) and (d). (f) and (g) HAADF images of representative local structural distortions in the microstructure. (h) HAADF image of an edge dislocation induced by local structural distortion. The strain distribution is shown in the inset image: $y = [2\bar{2}0]$ and $x = [00\bar{2}]$.

Additional simulation-based analyses were incorporated to better understand the configuration of Ga and Sb atoms co-occupying the vacant Pb sites. In Figure 3(c), the atomic-resolution HAADF image reveals lattice distortions resulting from GaSb co-occupying the Pb site. This observation is based on two key features: atomic radii and

contrast. In pristine PbSe, when viewed along the [110] zone axis, the cation and anion sites separate into distinct alternating atomic columns, leading to alternating bright (Pb columns, $Z = 82$) and dark contrast (Se columns, $Z = 34$). Due to the ordered occupation, the radii of the bright and dark columns remain uniform and do not vary spatially. These phenomena are corroborated by multislice simulations performed on pristine PbSe, as shown in Figure 3(d) and Figure S3(b).

When the Ga and Sb atoms co-occupied the vacant Pb sites, two representative and simplified configurations were considered to analyze how co-doping altered the HAADF image features. In the first case, for GaSb co-doping, Ga occupies the deficient Pb site, whereas Sb cannot fit within the site and shifts by $1/3$ of the unit cell along the [010] direction. The corresponding atomic model is presented as Atomic Model 1 in Figure S5. To emphasize the contrast changes, we assume this co-occupation occurs consistently along the z-direction. The multislice simulation suggests that the local lattice distortion closely resembles the experimental observations, as shown in Figure S3(a) and Figure S4(a), with both highlighted by a pair of red arrows. The line profiles indicate a drop in intensity at the cation sites for co-doping. This reduction is due to the significantly lower atomic number of Ga ($Z = 31$) compared to Pb ($Z = 82$). Additionally, the projected cation column radii decrease in the regions for co-doping, as the atomic radius of Ga ($r = 130$ pm) is smaller than that of Pb ($r = 180$ pm). Furthermore, in the line profile along the [010] direction, an additional atomic column overlapping with the Se columns is observed in the simulation, as shown in Figure S4(a). This is because Sb cannot fit within the Pb site alongside Ga, causing Sb columns to shift. The simulated results agree well with the experimental observations, as shown in Figure S3(a), where an additional atomic column overlaps with the Se column.

In the second case, Sb occupies the vacant Pb site, while Ga shifts by $1/3$ of the unit cell along the [010] direction. The corresponding atomic model is shown as Atomic Model 2 in Figure S5. We assume that this co-occupation extends uniformly along the z-direction. The multislice simulation suggests that the local lattice distortion is consistent with experimental observations, as illustrated in Figure S3(a) and Figure S4(b), with both highlighted by a pair of red arrows. The line profiles again indicate an

intensity drop at the cation sites, due to the lower atomic number of Sb ($Z = 51$) than Pb ($Z = 82$). Similar to the Atomic Model 1, the projected cation column radii decrease for co-doping, because Sb (atomic radius = 145 pm) is smaller than Pb (180 pm). Additionally, the line profile along the [010] direction reveals an extra atomic column overlapping with the Se columns, as shown in Figure S4(b). This phenomenon is because Ga cannot co-occupy the Pb site with Sb, leading to its displacement. The simulation results align well with experimental observations in Figure S3(a), where an additional atomic column overlaps with the Se column.

In summary, when GaSb co-occupies the deficient Pb site, the local image intensity of the cationic site and the radius of the atomic column decrease. Additionally, if the second atom in the GaSb pair cannot fit within the Pb site, an extra atomic column may become visible. While the precise image characteristics depend on the specific orientation of GaSb, the overall features should align with the trends discussed here. The experimentally observed local lattice distortion follows these predicted changes, further supporting the GaSb co-occupation at the deficient Pb site.

Figure S3. (a) Experimental atomic resolution HAADF image of $\text{Pb}_{0.99875}(\text{GaSb})_{0.00125}\text{Se}$ and (b) multislice simulation of pristine PbSe. The atomic model used for simulation is shown in Figure S5.

Figure S4. Multislice simulation of GaSb co-occupation with (a) Ga atom residing at the Pb site, and (b) Sb atom residing at the Pb site. The atomic models used for simulation are shown in Figure S5.

Figure S5. Atomic model for PbSe and $Pb_{1-x}(GaSb)_xSe$ with different configurations used for multislice simulation of HAADF images as shown in Figure S3 and Figure S4.

11. The authors state that the dynamic doping effect of Cu ions generates extra charge carriers, as Cu ions go into interstitial sites. However, the electrical conductivity decreases with Cu addition in most samples. This contradiction needs to be addressed. Authors' response: As shown in Figure 9(a), the electrical conductivity of all Cu addition samples increases when the temperature is above 573 K. This is mainly

because of the dynamic doping effect. As shown in Figure R3, the n_H does not increase at low temperatures; instead, it is lower than that of samples without interstitial Cu atoms. This may be because Cu atoms are more inclined to occupy the Pb vacancy and act as the acceptor at low temperatures, reducing the n_H . With the increase in temperature, the dynamic doping effect is obvious, resulting in the increase of the n_H .

Figure 9. Thermoelectric properties as a function of temperature for the $\text{Pb}_{0.99875}(\text{GaSb})_{0.00125}\text{Se}-y\text{Cu}$ ($y = 0, 0.1\%, 0.3\%, 0.5\%, 0.7\%, \text{ and } 0.9\%$) samples: (a) Electrical conductivity, σ ; (b) Seebeck coefficient, S ; (c) power factor, PF ; (d) total (κ_{tot}) and (e) lattice (κ_{lat}) thermal conductivities; and (f) figure of merit, ZT .

Figure R3. Carrier concentration as a function of temperature for the $\text{Pb}_{0.99875}(\text{GaSb})_{0.00125}\text{Se}-y\text{Cu}$ ($y = 0, 0.1\%, 0.3\%, 0.5\%, 0.7\%, \text{ and } 0.9\%$).

12. The authors should provide a more detailed description of each step in their synthesis method for clarity and reproducibility.

Authors' response: In the revised manuscript, we describe the synthesis details of each step in more detail. The sentences of: "The tubes containing the raw materials were flame-sealed at a pressure of approximately 2×10^{-3} torr, heated to 1173 K for 10 h, and then soaked at this temperature for 5 h. Finally, the samples were cooled to room temperature for 12 hours. The synthesized GaSb ingots will act as dopants." "For a typical sample, the following amounts of the raw materials were used: Pb (8 g, 38.61 mmol), GaSb (0.0093 g, 0.04857 mmol), Se (3.083 g, 39.04 mmol), Cu (0.0074 g, 0.11645 mmol), and Zn (0.0253 g, 0.3868 mmol) for preparing ~11 g ingot sample of $\text{Pb}_{0.99875}(\text{GaSb})_{0.00125}\text{Zn}_{0.01}\text{Se}_{1.01}-0.3\%\text{Cu}$." were added in methods part in the revised manuscript.

13. The authors should clarify the superlattice structure of GaSb-doped PbSe used for the band structure calculations.

Authors' response: Thanks for your comments. The $3 \times 3 \times 3$ supercell based on the rock-salt primitive cell of PbSe was adopted to evaluate the electronic structure of pure ($\text{Pb}_{27}\text{Se}_{27}$). In the manuscript, a Pb site in $\text{Pb}_{27}\text{Se}_{27}$ was replaced by a Ga atom and a Sb atom to calculate the effect of GaSb doping on the band structure of PbSe. Moreover, the band structure of the $5 \times 5 \times 5$ supercell of $\text{Pb}_{124}\text{GaSbSe}_{125}$ is calculated for closing

the experimental value.

Reviewer #2 (Remarks to the Author):

In this paper, the authors report that the thermoelectric performance of n-PbSe was improved by doping GaSb compounds due to the effects of band convergence and lattice distortion.

Band convergence and lattice distortion are now well-known strategies to improve thermoelectric performance in thermoelectric materials research, and many similar studies have been performed on similar PbTe materials up to less than 10 years ago. Since then, the same focus has shifted from PbTe to similar SnTe and PbSe. In particular, several papers on performance enhancement of PbSe have been published very recently (2023-2024) (see, for example, R1-R5), where values equivalent to the highest achieved thermoelectric performance in this paper have already been reported. Therefore, this paper does not have much advantage in thermoelectric performance improvement.

Authors' response: Band convergence plays an important role in improving the electrical properties of thermoelectric materials, as reported in previous literature. For p-type PbSe materials, the band convergence greatly improves the thermoelectric properties. For example, Y. Zhu et al. achieved multiple valence bands convergence in p-type PbSe by incorporating AgInSe₂, achieving an exceptional ZT of ~ 1.9 at 873 K in p-type PbSe.¹⁵ For n-type PbSe, the large energy difference (ΔE_c) between the first (L) and second (Σ) conduction bands poses a significant challenge for achieving conduction band convergence. Meanwhile, the performance of n-type PbSe is lower than that of p-type PbSe, which limits the construction of thermoelectric devices. In this work, the **conduction band convergence is realized** via GaSb-doping, resulting in large average PF and average ZT . Due to the lattice distortion caused by GaSb-doping, the point defect scattering is enhanced, and the κ_{lat} is significantly reduced. At the same time, the carrier mobility is almost not adversely affected.

- r5. Zhu, Y. et al. Multiple valence bands convergence and strong phonon scattering lead to high thermoelectric performance in p-type PbSe. *Nat. Commun.* **13**, 4179 (2022).

However, this paper might have a novel feature regarding the crystal structure of the samples, apart from other papers. The authors claim that two atoms, one Ga and one Sb, are substituted for one Pb deficiency. If this is true, it is very exciting regarding materials science research. The authors are trying to prove the substitution by microstructure observation and electronic structure calculation. However, I could not understand the proof. More details below. In the end, I do not recommend the publication of this paper.

Authors' response: Thank you for your recognition of our novel feature that one Ga and one Sb are substituted for one Pb deficiency. We are sorry for the unclear explanation of the co-occupation configuration through microstructure observation and electronic structure calculation in the manuscript. The new proofs and the explanation are highlighted in the manuscript with a yellow background as follows.

First of all, through the first-principles calculations, multiple possible configurations were evaluated. The different configurations of GaSb-substituted PbSe ($\text{GaSbPb}_{26}\text{Se}_{27}$) without structural relaxation are shown in Figures S1(a-e). Specifically, Figure S1(a) illustrates that Ga and Sb are substituted for one Pb atom in PbSe (configuration 1). Figures S1(b) and S1(c) depict that Ga substitutes one Pb atom while the Sb atom is positioned in an interstitial site close to (Figure S1(b), configuration 2) or far from (Figure S1(c), configuration 3) the Ga atom. Additionally, Figures S1(d) and S1(e) show that Sb substitutes one Pb atom while the Ga atom is located in an interstitial site close to (Figure S1(d), configuration 4) or far from (Figure S1(e), configuration 5) the Sb atom. Figures S1(f-j) are the corresponding configurations of Figures S1(a-e), with all ionic positions relaxed to equilibrium until the calculated Hellmann-Feynman forces on each atom were less than 0.02 eV/\AA . After relaxation, we observed that configuration 1, configuration 2, and configuration 4 exhibit similar Ga and Sb substitution configurations at one Pb site, with a total ground-state energy of -223.48 eV . In contrast, the total ground-state energies for configuration 3 and

configuration 5 are -222.65 and -222.61 eV, respectively. The energy differences indicate a preference for Ga and Sb to be located at one Pb site, as in configuration 1.

Figure S1. The possible configurations for GaSb-doped PbSe: (a) configuration 1, Ga and Sb substitute for one Pb atom; configuration 2 and configuration 3, Ga substitutes one Pb atom while the Sb atom is positioned in an interstitial site close to (b) or far from (c) the Ga atom; configuration 4 and configuration 5, Sb substitutes one Pb atom

while the Ga atom is located in an interstitial site close to (d) or far from (e) the Sb atom and (f-j) are the corresponding optimize configurations of (a-e).

Secondly, as shown in Figure 1(b), the lattice parameters of the GaSb-doped PbSe increased with increasing GaSb amount. Since the atomic radii of Ga and Sb are smaller than Pb, this can be attributed to the replacement of Pb with Ga and Sb.

Figure 1. (a) PXRD patterns and (b) refined lattice parameters of the $\text{Pb}_{1-x}(\text{GaSb})_x\text{Se}$ ($x = 0, 0.05\%, 0.075\%, 0.1\%, 0.125\%, 0.15\%$, and 0.175%) samples as a function of the GaSb content.

In addition, by analyzing the atomic-resolution HAADF images, we found that the intensity line profiles from the experimental results showed additional atoms corresponding to Ga ($Z = 31$) and Sb ($Z = 51$) near the Pb sites, which exhibited considerably low intensities than Pb in the HAADF image (Figure 3e). This local structural distortion could be caused by the fact that the Ga and Sb atoms co-occupy the vacant Pb sites, as predicted by the DFT calculations.

Figure 3. S/TEM analysis of the microstructure of the $\text{Pb}_{0.99875}(\text{GaSb})_{0.00125}\text{Se}$ sample. (a) HAADF image of a representative region with the corresponding EDS maps, where all elements are homogeneously distributed. Weak signals of Ga and Sb are observed owing to a low doping concentration. (b) SAED pattern along the $[110]$ zone axis acquired from the region shown in (a). (c) Atomic-resolution HAADF image showing the local structural distortion. (d) Multislice HAADF image simulation of pure PbSe without structural distortion. (e) Intensity line profiles of the regions highlighted in (c) and (d). (f) and (g) HAADF images of representative local structural distortions in the microstructure. (h) HAADF image of an edge dislocation induced by local structural distortion. The strain distribution is shown in the inset image: $y = [2\bar{2}0]$ and $x = [00\bar{2}]$.

Additional simulation-based analyses were incorporated to better understand the configuration of Ga and Sb atoms co-occupying the vacant Pb sites. In Figure 3(c), the atomic-resolution HAADF image reveals lattice distortions resulting from GaSb co-occupying the Pb site. This observation is based on two key features: atomic radii and

contrast. In pristine PbSe, when viewed along the [110] zone axis, the cation and anion sites separate into distinct alternating atomic columns, leading to alternating bright (Pb columns, $Z = 82$) and dark contrast (Se columns, $Z = 34$). Due to the ordered occupation, the radii of the bright and dark columns remain uniform and do not vary spatially. These phenomena are corroborated by multislice simulations performed on pristine PbSe, as shown in Figure 3(d) and Figure S3(b).

When the Ga and Sb atoms co-occupied the vacant Pb sites, two representative and simplified configurations were considered to analyze how co-doping altered the HAADF image features. In the first case, for GaSb co-doping, Ga occupies the deficient Pb site, whereas Sb cannot fit within the site and shifts by $1/3$ of the unit cell along the [010] direction. The corresponding atomic model is presented as Atomic Model 1 in Figure S5. To emphasize the contrast changes, we assume this co-occupation occurs consistently along the z-direction. The multislice simulation suggests that the local lattice distortion closely resembles the experimental observations, as shown in Figure S3(a) and Figure S4(a), with both highlighted by a pair of red arrows. The line profiles indicate a drop in intensity at the cation sites for co-doping. This reduction is due to the significantly lower atomic number of Ga ($Z = 31$) compared to Pb ($Z = 82$). Additionally, the projected cation column radii decrease in the regions for co-doping, as the atomic radius of Ga ($r = 130$ pm) is smaller than that of Pb ($r = 180$ pm). Furthermore, in the line profile along the [010] direction, an additional atomic column overlapping with the Se columns is observed in the simulation, as shown in Figure S4(a). This is because Sb cannot fit within the Pb site alongside Ga, causing Sb columns to shift. The simulated results agree well with the experimental observations, as shown in Figure S3(a), where an additional atomic column overlaps with the Se column.

In the second case, Sb occupies the vacant Pb site, while Ga shifts by $1/3$ of the unit cell along the [010] direction. The corresponding atomic model is shown as Atomic Model 2 in Figure S5. We assume that this co-occupation extends uniformly along the z-direction. The multislice simulation suggests that the local lattice distortion is consistent with experimental observations, as illustrated in Figure S3(a) and Figure S4(b), with both highlighted by a pair of red arrows. The line profiles again indicate an

intensity drop at the cation sites, due to the lower atomic number of Sb ($Z = 51$) than Pb ($Z = 82$). Similar to the Atomic Model 1, the projected cation column radii decrease for co-doping, because Sb (atomic radius = 145 pm) is smaller than Pb (180 pm). Additionally, the line profile along the [010] direction reveals an extra atomic column overlapping with the Se columns, as shown in Figure S4(b). This phenomenon is because Ga cannot co-occupy the Pb site with Sb, leading to its displacement. The simulation results align well with experimental observations in Figure S3(a), where an additional atomic column overlaps with the Se column.

In summary, when GaSb co-occupies the deficient Pb site, the local image intensity of the cationic site and the radius of the atomic column decrease. Additionally, if the second atom in the GaSb pair cannot fit within the Pb site, an extra atomic column may become visible. While the precise image characteristics depend on the specific orientation of GaSb, the overall features should align with the trends discussed here. The experimentally observed local lattice distortion follows these predicted changes, further supporting the GaSb co-occupation at the deficient Pb site.

Figure S3. (a) Experimental atomic resolution HAADF image of $\text{Pb}_{0.99875}(\text{GaSb})_{0.00125}\text{Se}$ and (b) multislice simulation of pristine PbSe. The atomic model used for simulation is shown in Figure S5.

Figure S4. Multislice simulation of GaSb co-occupation with (a) Ga atom residing at the Pb site, and (b) Sb atom residing at the Pb site. The atomic models used for simulation are shown in Figure S5.

Figure S5. Atomic model for PbSe and $\text{Pb}_{1-x}(\text{GaSb})_x\text{Se}$ with different configurations used for multislice simulation of HAADF images as shown in Figure S3 and Figure S4.

1. Band structure calculation

The band structure of PbSe in Fig. 6 is quite different from those in other papers (R6, R7), which were calculated by the same VASP program. I understand that the band structure depends somewhat on the calculation program. However, in the study of band

convergence using band edge structures, the shapes of the band structures seem to have a large influence.

R6) K. Hummer et al., Phys. Rev. B 75(2007)195211, "Structural and electronic properties of lead chalcogenides from first principles":

In Fig.1, an energy difference is 3 eV between L and Gamma points of CBM for PbSe, which was calculated by VASP.

R7) Y. Zhang et al., Phys. Rev. B 80(2009)024304, "Thermodynamic properties of PbTe, PbSe, and PbS: First-principles study":

In Fig.1(b), an energy difference is 0.9 eV between L point and Sigma axis of CBM for PbSe, which was calculated by VASP.

Authors' response: Thanks for your comments. The band structure of PbSe in the manuscript is indeed different from that of other papers (R6 and R7), which were calculated using the same VASP program. This difference is mainly due to the use of the supercell model in the manuscript. In papers (R6 and R7), the authors use the unit cell model. Using the supercell model causes the band to fold, which significantly affects the shapes of the band structure. The supercell model of $Pb_{27}Se_{27}$ has been widely used in the calculation of PbSe band structure. Such as H. Wang et al., Energy Environ. Sci. 7, 804–811 (2014). "Tuning bands of PbSe for better thermoelectric efficiency". The article mentions: *However, new evidence and data interpretation have indicated that the actual convergence temperature should be higher. In PbSe, the Σ band is further away (0.3 eV at 0 K) from the primary band maximum and the two bands converge at a higher temperature.* Such as M. Hong et al., J. Am. Chem. Soc. 142, 2672-2681 (2020). "Establishing the Golden Range of Seebeck Coefficient for Maximizing Thermoelectric Performance". The article mentions that *Figure 7a illustrates the calculated band structure of $Pb_{27}Se_{27}$, revealing E_g to be 0.25 eV, consistent with the reported value for PbSe.* The band structure and E_g reported in these papers are consistent with our calculated results.

The method of determining the atomic configuration used in the band calculations, as described in lines 439-458 of the main text and Fig. S1, is acceptable. However, there is no discussion of the stability of $\text{Pb}_{26}\text{GaSbSe}_{27}$. For example, its stability could be discussed by the formation energies of $\text{Pb}_{27}\text{Se}_{27}$, $\text{Pb}_{26}\text{GaSbSe}_{27}$, GaSb, and Pb.

Authors' response: Before determining the use of $\text{Pb}_{26}\text{GaSbSe}_{27}$ to calculate the band structure, we first evaluated multiple possible configurations to replace the Pb atom. Five configurations are considered, respectively. Ga and Sb are substituted for one Pb atom in PbSe (configuration 1). Ga substitutes one Pb atom while the Sb atom is positioned in an interstitial site close to (configuration 2) or far from (configuration 3) the Ga atom. Sb substitutes one Pb atom while the Ga atom is located in an interstitial site close to (configuration 4) or far from (configuration 5) the Sb atom. By calculating the formation energy of different configurations, the one Ga and one Sb atom co-occupy a Pb position, showing the lowest formation energy. The total ground state energy is -223.48 eV, leading to the highest stability of the structures.

The chemical composition of the sample is, for example, $\text{Pb}_{0.99875}(\text{GaSb})_{0.00125}\text{Se}$, whereas the chemical composition of the band calculation target is $\text{Pb}_{26}(\text{GaSb})\text{Se}_{27}$. The GaSb content differs by a factor of about 30 between them. Is this band calculation result valid for the discussion of sample properties?

Authors' response: Thanks for your comments. Due to the low doping amount of GaSb, it is a challenge to expand the cell to $\text{Pb}_{799}(\text{GaSb})_1\text{Se}_{800}$ for theoretical calculations. Therefore, we calculate the band structure of the sample, which is close to the experimental value. Figure S10 shows the band structure of $\text{Pb}_{124}(\text{GaSb})_1\text{Se}_{125}$. GaSb-doping decreased the ΔE_c between the Σ and L bands from 0.33 eV for the intrinsic PbSe to 0.13 eV for the GaSb-doped PbSe. Figure 6c shows the band structure of $\text{Pb}_{26}(\text{GaSb})_1\text{Se}_{27}$. GaSb-doping decreased the ΔE_c between the Σ and L bands from 0.33 eV for the intrinsic PbSe to 0.08 eV for the GaSb-doped sample. According to the theoretical calculation results, the ΔE_c value is reduced with the GaSb content. Therefore, GaSb-doping can realize the conduction band convergence.

Figure S10. Electronic band structures for GaSb-doped PbSe ($\text{Pb}_{124}\text{GaSbSe}_{125}$).

Figure 6. (a, c) Electronic band structures for the intrinsic and GaSb-doped PbSe-based material, respectively. (b, d) Projected DOS for the pure and GaSb-doped PbSe-based material, respectively.

2. Microstructural analysis

I could not understand from Fig. 3 alone that GaSb is replacing Pb deficiency, although I understand that there is lattice distortion.

The interior of the sample appears to be locally Ga/Sb doped, rather than GaSb doped throughout. In other words, it appears to be a two-phase or composite structure, where the parent phase of the first phase is PbSe and the second phase is Ga/Sb-doped PbSe

with a distorted lattice.

If so, the presence of the second phase may have the effect of reducing thermal conductivity. However, the thermoelectric properties would be roughly the volume average of the thermoelectric properties of each phase, would they not? However, since the volume fraction of phase 2 is small, it may not have much effect.

Authors' response: The additional simulation-based analysis of TEM results and detailed discussion are added to better understand the experimental observations in the revised manuscript.

In Figures 3c, f, and g, the atomic-resolution HAADF image reveals lattice distortions resulting from GaSb co-occupying the Pb site. This observation is based on two key features: atomic radii and contrast. In pristine PbSe, when viewed along the [110] zone axis, the cation and anion sites separate into distinct alternating atomic columns, leading to alternating bright (Pb columns, $Z = 82$) and dark contrast (Se columns, $Z = 34$). Due to the ordered occupation, the radii of the bright and dark columns remain uniform and do not vary spatially. These phenomena are corroborated by multislice simulations performed on pristine PbSe, as shown in Figure 3d and Figure S3(b).

When the Ga and Sb atoms co-occupied the vacant Pb sites, two representative and simplified configurations were considered to analyze how co-doping altered the HAADF image features. In the first case, for GaSb co-doping, Ga occupies the deficient Pb site, whereas Sb cannot fit within the site and shifts by $1/3$ of the unit cell along the [010] direction. The corresponding atomic model is presented as Atomic Model 1 in Figure S5. To emphasize the contrast changes, we assume this co-occupation occurs consistently along the z-direction. The multislice simulation suggests that the local lattice distortion closely resembles the experimental observations, as shown in Figure S3(a) and Figure S4(a), with both highlighted by a pair of red arrows. The line profiles indicate a drop in intensity at the cation sites for co-doping. This reduction is due to the significantly lower atomic number of Ga ($Z = 31$) compared to Pb ($Z = 82$). Additionally, the projected cation column radii decrease in the regions for co-doping, as the atomic radius of Ga ($r = 130$ pm) is smaller than that of Pb ($r = 180$ pm). Furthermore, in the

line profile along the [010] direction, an additional atomic column overlapping with the Se columns is observed in the simulation, as shown in Figure S4(a). This is because Sb cannot fit within the Pb site alongside Ga, causing Sb columns to shift. The simulated results agree well with the experimental observations, as shown in Figure S3(a), where an additional atomic column overlaps with the Se column.

In the second case, Sb occupies the vacant Pb site, while Ga shifts by 1/3 of the unit cell along the [010] direction. The corresponding atomic model is shown as Atomic Model 2 in Figure S5. We assume that this co-occupation extends uniformly along the z-direction. The multislice simulation suggests that the local lattice distortion is consistent with experimental observations, as illustrated in Figure S3(a) and Figure S4(b), with both highlighted by a pair of red arrows. The line profiles again indicate an intensity drop at the cation sites, due to the lower atomic number of Sb ($Z = 51$) than Pb ($Z = 82$). Similar to the Atomic Model 1, the projected cation column radii decrease for co-doping, because Sb (atomic radius = 145 pm) is smaller than Pb (180 pm). Additionally, the line profile along the [010] direction reveals an extra atomic column overlapping with the Se columns, as shown in Figure S4(b). This phenomenon is because Ga cannot co-occupy the Pb site with Sb, leading to its displacement. The simulation results align well with experimental observations in Figure S3(a), where an additional atomic column overlaps with the Se column.

In summary, when GaSb co-occupies the deficient Pb site, the local image intensity of the cationic site and the radius of the atomic column decrease. Additionally, if the second atom in the GaSb pair cannot fit within the Pb site, an extra atomic column may become visible. While the precise image characteristics depend on the specific orientation of GaSb, the overall features should align with the trends discussed here. The experimentally observed local lattice distortion follows these predicted changes, further supporting the GaSb co-occupation at the deficient Pb site.

Figure S3. (a) Experimental atomic resolution HAADF image of $\text{Pb}_{0.99875}(\text{GaSb})_{0.00125}\text{Se}$ and (b) multislice simulation of pristine PbSe. The atomic model used for simulation is shown in Figure S5.

Figure S4. Multislice simulation of GaSb co-occupation with (a) Ga atom residing at the Pb site, and (b) Sb atom residing at the Pb site. The atomic models used for simulation are shown in Figure S5.

Figure S5. Atomic model for PbSe and $\text{Pb}_{1-x}(\text{GaSb})_x\text{Se}$ with different configurations used for multislice simulation of HAADF images as shown in Figure S3 and Figure S4.

This localized doping behavior arises due to the low concentration of the dopant in $\text{Pb}_{0.99875}(\text{GaSb})_{0.00125}\text{Se}$. Given the small amount of dopant introduced, it does not uniformly expand across the entire material matrix but instead remains confined to specific local regions. The similar phenomenon has been observed in well-acknowledged Cu interstitial formation in PbSe and PbTe due to low concentration.^{r6-r9}

Therefore, we do not consider this to be a two-phase or composite structure. Additionally, we conducted structural analysis across multiple length scales, none of which indicate phase segregation. Instead, the results suggest a uniform distribution of GaSb dopants within the material.

The PXRD patterns of the $\text{Pb}_{1-x}(\text{GaSb})_x\text{Se}$ ($x = 0, 0.05\%, 0.075\%, 0.1\%, 0.125\%, 0.15\%$, and 0.175%) samples shown in Figure 1(a) confirm that all samples are single-phase and are crystallized in the rock-salt structure with the $Fm\bar{3}m$ space group (PDF#06-0354).

Figure 1. (a) PXRD patterns and (b) refined lattice parameters of the $\text{Pb}_{1-x}(\text{GaSb})_x\text{Se}$ ($x = 0, 0.05\%, 0.075\%, 0.1\%, 0.125\%, 0.15\%$, and 0.175%) samples as a function of the GaSb content.

Figure 2(a) shows the backscattered electron (BSE) SEM image, which exhibits a uniform contrast, indicating the absence of any noticeable second-phase region. Figure 2(b) shows the energy dispersive spectroscopy (EDS) results, demonstrating that Pb, Se, Ga, and Sb are evenly distributed in $\text{Pb}_{0.99875}(\text{GaSb})_{0.00125}\text{Se}$.

Figure 2. Microstructure and composition analysis of the $\text{Pb}_{0.99875}(\text{GaSb})_{0.00125}\text{Se}$ sample. (a) SEM-BSE image of the specimen and (b) EDS mapping of the region shown in (a).

Figure 3(a) shows the high-angle annular dark-field (HAADF) image of a representative region of the $\text{Pb}_{0.99875}(\text{GaSb})_{0.00125}\text{Se}$ sample. The uniform contrast in the low-magnification HAADF image indicates the absence of phase segregation or precipitate formation. This observation is supported by EDS, which revealed a uniform distribution of the elemental species. Notably, the GaSb doping concentration was low and close to the detection limit, which resulted in weak EDS signals for Ga and Sb. In addition, the selected area electron diffraction (SAED) pattern (Figure 3b) along the [110] axis from the region shown in Figure 3(a) matches the rock-salt structure with the space group $Fm\bar{3}m$. No additional diffraction spots or streaking was observed in the SAED pattern, suggesting the solid solution of the sample.

Figure 3. S/TEM analysis of the microstructure of the $\text{Pb}_{0.99875}(\text{GaSb})_{0.00125}\text{Se}$ sample. (a) HAADF image of a representative region with the corresponding EDS maps, where all elements are homogeneously distributed. Weak signals of Ga and Sb are observed owing to a low doping concentration. (b) SAED pattern along the $[110]$ zone axis acquired from the region shown in (a). (c) Atomic-resolution HAADF image showing the local structural distortion. (d) Multislice HAADF image simulation of pure PbSe without structural distortion. (e) Intensity line profiles of the regions highlighted in (c) and (d). (f) and (g) HAADF images of representative local structural distortions in the microstructure. (h) HAADF image of an edge dislocation induced by local structural distortion. The strain distribution is shown in the inset image: $y = [2\bar{2}0]$ and $x = [00\bar{2}]$.

- r6. Zhou, C. et al. High-Performance n-Type PbSe – Cu₂Se Thermoelectrics through Conduction Band Engineering and Phonon Softening. *J. Am. Chem. Soc.* **140**, 15535-15545 (2018).
- r7. Xu, L. et al. Enhanced average power factor and ZT value in PbSe thermoelectric material with dual interstitial doping. *Energy Environ. Sci.* **17**, 2018-2027 (2024).

- r8. Xiao, Y. et al. Remarkable Roles of Cu To Synergistically Optimize Phonon and Carrier Transport in n-Type PbTe-Cu₂Te. *J. Am. Chem. Soc.* **139**, 18732-18738 (2017).
- r9. Xiao, Y. et al. Cu Interstitials Enable Carriers and Dislocations for Thermoelectric Enhancements in n-PbTe_{0.75}Se_{0.25}. *Chem* **6**, 523-537 (2020).

R1) B. Ge et al., *Energy Environ. Sci.* 16(2023)3994, "Engineering an atomic-level crystal lattice and electronic band structure for an extraordinarily high average thermoelectric figure of merit in n-type PbSe":

The abstract says "the best composition achieves a remarkably high average power factor of $24 \mu\text{W cm}^{-1} \text{K}^{-2}$ from 300 to 823 K, with a substantially depressed lattice thermal conductivity of $0.2 \text{ W m}^{-1} \text{K}^{-1}$ at 723 K. With a ZT of 0.55 at 300 K, an average ZT is 1.30 from 400 to 823 K".

R2) J. Cai et al., *Adv. Funct. Mater.* 34(2024)2311217, "Giant Band Convergence and High Thermoelectric Performance in n-Type PbSe Induced by Spin-Orbit Coupling":

The abstract says "Owing to the achieved giant band convergence and the low thermal conductivity, a high peak zT of 1.75 at 850 K and an outstanding average zT of 1.04 are obtained."

R3) D. Zhang et al., *Scripta Materialia*, 244(2024)116003, "High thermoelectric performance of PbSe via a synergistic band engineering and dislocation approach":

The abstract says "a high average ZT_{ave} 0.97 between 300 and 873 K is obtained for Pb_{0.9985}Gd_{0.0015}Se-0.2 %Cu."

R4) S. Wang et al., *Energy Environ. Sci.* 17(2024)2588, "Realizing high-performance thermoelectric modules through enhancing the power factor via optimizing the carrier mobility in n-type PbSe crystals":

The abstract says "the ZT values of the Pb_{1.006}Se+0.0016Al crystal reached 0.5 at 300 K, 1.5 at 673 K, and the average ZT (ZT_{ave}) reached 1.1 at 300-773 K."

R5) S. Wu et al., J. Mater. Chem. A 12(2024)26013, "Optimizing the thermoelectric performance of n-type PbSe through dynamic doping driven by entropy engineering": The abstract says "the thermoelectric properties of $\text{Pb}_{0.875}\text{Sn}_{0.125}\text{Se}_{0.5}\text{Te}_{0.25}\text{S}_{0.25}\text{-2at\%Cu}$ are remarkably improved, with a maximum dimensionless merit ZT_{max} of 1.46 at 623 K and an average dimensionless merit ZT_{ave} of 1.15 at 300-700 K."

Authors' response: Achieving high-stability thermoelectric materials with an excellent average power factor (PF_{avg}) and figure of merit (ZT_{avg}) is crucial for maximizing the output power density (ω) and conversion efficiency (η) of thermoelectric devices. Compared with p-type PbSe, the large energy difference (ΔE_c) between the Σ and L conduction bands poses a significant challenge for achieving conduction band convergence and a high PF .

Here, we have made a significant advance in raising the performance of n-type PbSe with GaSb semiconductor doping. We have obtained key new scientific insights on the role of GaSb in the PbSe structure, which will be influential in designing better thermoelectric materials. GaSb doping decreased the ΔE_c from 0.33 eV for the intrinsic PbSe to 0.08 eV for the GaSb-doped PbSe, realizing the conduction band convergence. Conduction band convergence leads to significantly increased degeneracy and Seebeck coefficient without deteriorating the electrical conductivity. Consequently, the $\text{Pb}_{0.99875}(\text{GaSb})_{0.00125}\text{Se}$ exhibited a high PF_{avg} of $\sim 22.37 \mu\text{W cm}^{-1} \text{K}^{-2}$ in the temperature range of 300–873 K, which is crucial for achieving high ω . At the same time, unlike traditional single-element doping, first-principles theoretical studies and microstructural analysis find a striking new insight that Ga and Sb atoms co-occupied at the Pb sites, causing local structural distortion and reduced κ_{lat} . Finally, a high ZT_{avg} of ~ 0.94 for 300 to 873 K was obtained. Moreover, introducing interstitial Cu and discordant Zn atoms further reduced the κ_{lat} to $\sim 0.4 \text{ W m}^{-1} \text{K}^{-1}$ at 873 K, close to the limit of amorphous PbSe. As a result, $\text{Pb}_{0.99875}(\text{GaSb})_{0.00125}\text{Zn}_{0.01}\text{Se}_{1.01}\text{-0.3\%Cu}$ sample exhibits a record-high ZT_{avg} of ~ 1.01 in the temperature range of 300–873 K.

The novelty and significance of the GaSb-doped and interstitial Cu and discordant Zn atoms alloyed n-type PbSe thermoelectric materials are summarized below:

(i) Conduction band convergence is achieved in n-type PbSe thermoelectric materials. GaSb-doping flattened the conduction band and reduced the ΔE_c from 0.33 eV for PbSe to 0.08 eV for GaSb-doped PbSe, significantly increasing degeneracy and Seebeck coefficient.

(ii) First-principles theoretical studies and microstructural analysis indicate a striking new insight that the Ga and Sb atoms co-occupied at the Pb sites, causing local structural distortion and reducing the κ_{lat} . This is totally different from traditional single-element doping.

(iii) Interstitial Cu and discordant Zn atoms further reduced the κ_{lat} and an ultralow κ_{lat} value of $\sim 0.4 \text{ W m}^{-1} \text{ K}^{-1}$ at 873 K was obtained, close to the limit of amorphous PbSe.

Finally, the PF is improved by conduction band convergence and the κ_{lat} is reduced by interstitial Cu and discordant Zn atoms simultaneously. As a result, the $\text{Pb}_{0.99875}(\text{GaSb})_{0.00125}\text{Zn}_{0.01}\text{Se}_{1.01}\text{-0.3\%Cu}$ exhibits a record-high ZT_{avg} of ~ 1.01 in the temperature range of 300–873 K. We provide a new way to improve the ZT further, and it is expected to obtain high-performance n-type PbSe materials matching p-type PbSe materials.

We analyzed the advantages of our reported samples compared with papers R1-R5.

R1) B. Ge et al., Energy Environ. Sci. 16(2023)3994, "Engineering an atomic-level crystal lattice and electronic band structure for an extraordinarily high average thermoelectric figure of merit in n-type PbSe":

The abstract reported that "*the best composition achieves a remarkably high average power factor of $24 \mu\text{W cm}^{-1} \text{ K}^{-2}$ from 300 to 823 K, with a substantially depressed lattice thermal conductivity of $0.2 \text{ W m}^{-1} \text{ K}^{-1}$ at 723 K. With a ZT of 0.55 at 300 K, an average ZT is 1.30 from 400 to 823 K*". **In this paper, the high average ZT and average PF were reported for $\text{Cu}_{0.0029}\text{Pb}(\text{Se}_{0.8}\text{Te}_{0.2})_{0.95}$. The multi-level defect structures enhance the scattering of phonons and greatly reduce the κ_{lat} . The increased conduction band effective mass results in an increase in Seebeck coefficients.**

In our work, the high PF was obtained by the realization of conduction band

convergence. The $\text{Pb}_{0.99875}(\text{GaSb})_{0.00125}\text{Zn}_{0.01}\text{Se}_{1.01}-0.3\%\text{Cu}$ exhibits a record-high ZT_{avg} of ~ 1.01 in the temperature range of 300–873 K for Te-free PbSe-based materials, which is a significant advantage in terms of economic benefits. Our solid solution sample is beneficial in thermoelectric device preparation. Moreover, our sample has a wider operating temperature range, leading to good conversion efficiency and output power density over a wider temperature range.

R2) J. Cai et al., Adv. Funct. Mater. 34(2024)2311217, "Giant Band Convergence and High Thermoelectric Performance in n-Type PbSe Induced by Spin-Orbit Coupling":

The abstract reported that “*Owing to the achieved giant band convergence and the low thermal conductivity, a high peak zT of 1.75 at 850 K and an outstanding average zT of 1.04 are obtained*”. **The simultaneous achievement of a high average PF and average ZT is crucial for developing highly efficient devices. The average PF determines the actual output power of the thermoelectric device. Compared with p-type PbSe, the large energy difference (ΔE_c) between the Σ and L conduction bands poses a significant challenge for achieving conduction band convergence and a high PF . In our work, the high PF was obtained by the realization of conduction band convergence. In this paper, the maximum PF and average PF at 300-850K are only $15.5 \mu\text{W cm}^{-1} \text{K}^{-2}$ and $14.4 \mu\text{W cm}^{-1} \text{K}^{-2}$, much lower than our reported values.**

R3) D. Zhang et al., Scripta Materialia, 244(2024)116003, "High thermoelectric performance of PbSe via a synergistic band engineering and dislocation approach":

The abstract reported that “*a high average ZT_{ave} 0.97 between 300 and 873 K is obtained for $\text{Pb}_{0.9985}\text{Gd}_{0.0015}\text{Se}-0.2\%\text{Cu}$* ”. **This paper reports an average ZT close to ours, but the highest ZT is 1.4, which is lower than our reported value of 1.57. At the same time, it takes about 3 days to synthesize samples in this paper, while we synthesized polycrystalline samples in less than 1 day.**

R4) S. Wang et al., Energy Environ. Sci. 17(2024)2588, "Realizing high-performance thermoelectric modules through enhancing the power factor via optimizing the carrier mobility in n-type PbSe crystals":

The abstract reported that “*the ZT values of the $Pb_{1.006}Se+0.0016Al$ crystal reached 0.5 at 300 K, 1.5 at 673 K, and the average ZT (ZT_{ave}) reached 1.1 at 300-773 K*”. **This paper has obtained a high average ZT for n-type PbSe crystals. The mechanical properties of single crystal materials are often lower than those of polycrystalline materials. The preparation process of single crystal materials is complex and time-consuming, which is a challenge for large-scale production. At the same time, the polycrystalline sample we synthesized has a wider operating temperature range than this paper, leading to high conversion efficiency and output power density over a wider temperature range.**

R5) S. Wu et al., J. Mater. Chem. A 12(2024)26013, "Optimizing the thermoelectric performance of n-type PbSe through dynamic doping driven by entropy engineering":

The abstract reported that “*the thermoelectric properties of $Pb_{0.875}Sn_{0.125}Se_{0.5}Te_{0.25}S_{0.25-2at\%Cu}$ are remarkably improved, with a maximum dimensionless merit ZT_{max} of 1.46 at 623 K and an average dimensionless merit ZT_{ave} of 1.15 at 300-700 K*”. **In this paper, the κ_{tot} is greatly reduced through entropy engineering. However, due to the introduction of a large number of defects in entropy engineering, the μ_H is seriously reduced. The average PF in 300-673 K is only $17.8 \mu W cm^{-1} K^{-2}$. In our work, the sample we synthesized has a wider operating temperature range than this paper, leading to high conversion efficiency and output power density over a wider temperature range.**

Reviewer #3 (Remarks to the Author):

Article Title: "Conduction band convergence and local structure distortion for a superior thermoelectric performance of GaSb-doped n type PbSe thermoelectrics" The article reports the synthesis of the n-type PbSe solid solutions via GaSb doping resulting in superior TE properties. DFT calculations were also carried out to prove the high Seebeck coefficient because of band convergence. Thermal conductivity was further

reduced by addition of interstitial Cu atoms and Zn alloying as a result of which ZT values were enhanced upto 1.57. Results are interesting. Overall the manuscript have good introduction, experimental details are adequate. Discussion of results is very good. I have following questions/suggestions for revisions.

Authors' response: Thank you for your positive comments. We have carefully and significantly revised the manuscript.

1. As written in the manuscript $Pb_{1-x}(GaSb)_xSe$ ($x = 0, 0.05\%, 0.075\%, 0.1\%, 0.125\%, 0.15\%$, and 0.175%). Non-doped sample data is only provided in the XRD graphs. Authors are suggested to add the non-doped sample data for comparison in all graphs to account for the actual increment in TE properties with this doping.

Authors' response: Thanks for your comments. In the revised manuscript, the data of the non-doped PbSe sample was added.

Figure 4. (a) Temperature-dependent Hall coefficient, R_H ; (b) carrier concentration, n_H ; and (c) carrier mobility, μ_H for the $Pb_{1-x}(GaSb)_xSe$ ($x = 0, 0.05\%, 0.075\%, 0.1\%, 0.125\%, 0.15\%$, and 0.175%) samples. (d) μ_H as a function of n_H for n-type PbSe-based thermoelectrics.

Figure 5. Thermoelectric properties as a function of temperature for the $\text{Pb}_{1-x}(\text{GaSb})_x\text{Se}$ ($x = 0, 0.05\%, 0.075\%, 0.1\%, 0.125\%, 0.15\%,$ and 0.175%) samples: (a) Electrical conductivity, σ ; (b) Seebeck coefficient, S ; and (c) Seebeck coefficient as a function of n_H at room temperature. (d) Seebeck coefficient as a function of n_H at different temperatures (300, 373, 473, 573, 673, and 773 K) for the $\text{Pb}_{0.99875}(\text{GaSb})_{0.00125}\text{Se}$ sample.

Figure 7. Thermal conductivity as a function of temperature for the $\text{Pb}_{1-x}(\text{GaSb})_x\text{Se}$ ($x = 0, 0.05\%, 0.075\%, 0.1\%, 0.125\%, 0.15\%,$ and 0.175%) samples: (a) total (κ_{tot}) and (b) lattice (κ_{lat}) thermal conductivities.

Figure 8. Figure of merit (ZT) as a function of temperature for the $Pb_{1-x}(GaSb)_xSe$ ($x = 0, 0.05\%, 0.075\%, 0.1\%, 0.125\%, 0.15\%$, and 0.175%) samples.

Figure S11. Temperature-dependent (a) electronic thermal conductivity, κ_{ele} ; (b) thermal diffusivity, D ; (c) heat capacity, C_p ; and (d) Lorenz numbers, L for $Pb_{1-x}(GaSb)_xSe$ samples ($x = 0, 0.05\%, 0.075\%, 0.1\%, 0.125\%, 0.15\%$, and 0.175%).

2. Figure 6 shows band structure and density of states. Indicate the fermi level position in both graphs. If it is at zero then the explanation of bands crossing fermi level should also be added.

Authors' response: Thanks for your suggestions. The Fermi level position is at zero,

which has been marked in the revised Figure 6. The conduction band of GaSb-doped n-type PbSe crosses the Fermi level, mainly because the n_H increases due to the doping of GaSb. Add as follows: The corresponding description is given in the revised manuscript. The conduction band of GaSb-doped PbSe-based material crosses the Fermi level, mainly because the n_H increases due to the doping of GaSb, which makes the carrier fill the conduction band.

Figure 6. (a, c) Electronic band structures for the intrinsic and GaSb-doped PbSe-based material, respectively. (b, d) Projected DOS for the pure and GaSb-doped PbSe-based material, respectively.

3. Can authors elaborate more on the behavior of κ_{latt} upon addition of interstitial Cu atoms. Why the decrease was dominant in only above 573K.

Authors' response: This is mainly related to the dynamic doping of Cu atoms in PbSe. The change of electrical conductivity with temperature leads to a visible platform around 573 K, which is consistent with the literature reports.^{r10,r11} This is mainly because more Cu ions enter the PbSe lattice as the temperature increases, providing additional charge carriers. Similarly, the corresponding κ_{lat} shows a significant decrease at temperatures greater than 573 K. This can be attributed to the vibration and diffusion

of interstitial Cu atoms at high temperatures. This significantly enhances the scattering of phonons at high temperatures.

- r10. You, L. et al. Boosting the thermoelectric performance of PbSe through dynamic doping and hierarchical phonon scattering. *Energy Environ. Sci.* **11**, 1848-1858 (2018).
- r11. Zhou, C. et al. Exceptionally High Average Power Factor and Thermoelectric Figure of Merit in n-type PbSe by the Dual Incorporation of Cu and Te. *J. Am. Chem. Soc.* **142**, 15172-15186 (2020).

Reviewer #4 (Remarks to the Author):

Zhou, Jing. et al. demonstrated high thermoelectric figure of merit zT in PbSe by advanced strategies, including conduction band convergence and local structure distortion. High average zT (~ 1.01 in 300 to 873 K) was obtained in $\text{Pb}_{0.99875}(\text{GaSb})_{0.00125}\text{Se}-0.3\%\text{Cu}-1\%\text{ZnSe}$. The conclusions and findings in this study are strongly supported by experimental (XRD, SEM and TEM observation, electrical and transport properties measurements) and theoretical (DFT calculation of electronic structures) results. Therefore, this work should be of widespread interest to researchers in the fields of thermoelectrics and natural sciences. There is no major concern. I would recommend it for acceptance in its current form.

Authors' response: We appreciate your positive feedback and strong recommendation of our work for publication in Nature Communications.

Response Letter to Reviewers

Reviewer #2 (Remarks to the Author):

In the revised manuscript, the authors provide a more detailed microstructural analysis of the samples to prove that GaSb substitutes for the Pb site. This makes it more convincing. On the other hand, they do not discuss the stability of the samples using electronic structure calculation on the total energy, which I pointed out in the first review. That is, it is not clarified which is larger, the formation energy of $\text{Pb}_{26}(\text{GaSb})\text{Se}_{27}$ or the sum of the formation energies of 26PbSe , GaSb, and Se. Also, the authors explain the superiority of this paper over the previously published ones regarding thermoelectric performance improvement. The improvements seem to be small, but they exist.

I recommend the publication of this paper because this substitution is a new strategy to improve thermoelectric performance.

Authors' response: We appreciate your positive feedback on our work for publication in Nature Communications.

Thank you for your recognition of the view that GaSb substitutes for the Pb site. Through DFT calculation, the ground state energy of $\text{Pb}_{26}(\text{GaSb})\text{Se}_{27}$ ($E_{\text{Pb}_{26}(\text{GaSb})\text{Se}_{27}}$) is -223.48 eV. The ground state energies of 26PbSe ($E_{\text{Pb}_{26}\text{Se}_{26}}$), GaSb (E_{GaSb}), and Se (E_{Se}) are -213.85 eV, -7.34 eV, and -2.91 eV, respectively. The formation energy was calculated according to the Formula $\Delta H_{\text{AB}} = E_{\text{AB}} - E_{\text{A}} - E_{\text{B}}$. Here, ΔH_{AB} represents the formation energy of compound AB, E_{AB} represents the ground state energy of compound AB composed of elements A and B, E_{A} and E_{B} represent the ground state energy of elements A and B.^{†1} The calculated $\Delta H_{\text{Pb}_{26}(\text{GaSb})\text{Se}_{27}}$, $\Delta H_{\text{Pb}_{26}\text{Se}_{26}}$, ΔH_{GaSb} and ΔH_{Se} are respectively -56.471 eV, -46.514 eV, -0.58 eV and 0 eV. The formation energy of $\text{Pb}_{26}(\text{GaSb})\text{Se}_{27}$ ($= -56.471$ eV) is less than the sum of the formation energies of $\text{Pb}_{26}\text{Se}_{26}$, GaSb, and Se ($= -47.094$ eV). Therefore, the sample is more inclined to form $\text{Pb}_{26}(\text{GaSb})\text{Se}_{27}$ rather than decompose and precipitate. High-stability thermoelectric materials with an excellent average power factor (PF_{avg}) and figure of

merit (ZT_{avg}) have been obtained. That is crucial for maximizing the output power density (ω) and conversion efficiency (η) of thermoelectric devices. Moreover, thank you for your recognition of the improvement of thermoelectric performance.

r1. Feng Z, et al. Thermoelectric optimization of AgBiSe₂ by defect engineering for room-temperature applications. *Phy. Rev. B* **99**, 155203 (2019).

Reviewer #3 (Remarks to the Author):

The authors have revised the manuscript according to comments. Manuscript is now significantly improved and now I recommend it for publication.

Authors' response: We appreciate your positive feedback on our work for publication in Nature Communications.